# Shelf-Life Prediction and Critical Value of Quality Index of Sichuan Sauerkraut Based on Kinetic Model and Principal Component Analysis

**DOI:** 10.3390/foods11121762

**Published:** 2022-06-15

**Authors:** Jie Du, Min Zhang, Lihui Zhang, Chung Lim Law, Kun Liu

**Affiliations:** 1State Key Laboratory of Food Science and Technology, School of Food Science and Technology, Jiangnan University, Wuxi 214122, China; jiedu0803@outlook.com (J.D.); lihuizhang0205@163.com (L.Z.); 2International Joint Laboratory on Food Safety, Jiangnan University, Wuxi 214122, China; 3Jiangsu Province International Joint Laboratory on Fresh Food Smart Processing and Quality Monitoring, Jiangnan University, Wuxi 214122, China; 4Department of Chemical and Environmental Engineering, Malaysia Campus, University of Nottingham, Semenyih 43500, Selangor, Malaysia; chung-lim.law@nottingham.edu.my; 5Sichuan Tianwei Food Group Co., Ltd., Chengdu 610207, China; yfxm_liukun@teway.cn

**Keywords:** reaction kinetics model, arrhenius model, BP-ANN, total acid, E-tongue, GC-MS

## Abstract

Kinetic models and accelerated shelf-life testing were employed to estimate the shelf-life of Sichuan sauerkraut. The texture, color, total acid, microbe, near-infrared analysis, volatile components, taste, and sensory evaluation of Sichuan sauerkraut stored at 25, 35, and 45 °C were determined. Principal component analysis (PCA) and Fisher discriminant analysis (FDA) were used to analyze the e-tongue data. According to the above analysis, Sichuan sauerkraut with different storage times can be divided into three types: completely acceptable period, acceptable period, and unacceptable period. The model was found to be useful to determine the critical values of various quality indicators. Furthermore, the zero-order kinetic reaction model (R^2^, 0.8699–0.9895) was fitted better than the first-order kinetic reaction model. The Arrhenius model (*E_a_* value was 47.23–72.09 kJ/mol, *k_ref_* value was 1.076 × 10^6^–9.220 × 10^10^ d^−1^) exhibited a higher fitting degree than the Eyring model. Based on the analysis of physical properties, the shelf-life of Sichuan sauerkraut was more accurately predicted by the combination of the zero-order kinetic reaction model and the Arrhenius model, while the error back propagation artificial neural network (BP-ANN) model could better predict the chemical properties. It is a better choice for dealers and consumers to judge the shelf life and edibility of food by shelf-life model.

## 1. Introduction

Sichuan sauerkraut, a traditional fermented vegetable with a special flavor and beneficial active compounds, is popularly consumed in China [1]. Sauerkraut manufacturing is an important industry in the Sichuan province of China, with an annual output of five million tons worth more than 42 billion yuan (US $6.2 billion) in 2018 [2]. Sichuan sauerkraut is a food with a long shelf-life, up to 15 months. However, the texture, color and flavor of Sichuan sauerkraut will change greatly during the storage process. In the production process of Sichuan sauerkraut, heat sterilization technology was used to inactivate most of the microorganisms. The addition of potassium sorbate and sodium D-isoascorbate inhibited the growth of lactic acid bacteria and molds. Therefore, microorganisms can not be used as a standard to judge the shelf-life of modified products. From the perspective of consumers, it is more reasonable to use flavor and sensory indicators to determine the shelf life [3]. In practical applications, it is of great significance to establish a link between the flavor and sensory indicators and the shelf-life model. For distributors and consumers, the shelf-life and best before date of food can be obtained more quickly.

E-tongue is a powerful tool for distinguishing flavor characteristics, allowing for subtle differences in the taste of foods from different sources without regard to subjective factors [4,5]. It is defined as “a multi-sensory system consisting of multiple low-selectivity sensors and signal processing using advanced mathematical programs based on pattern recognition and multivariate data analysis” [6,7]. PCA and FDA can classify the electronic tongue data of Sichuan sauerkraut at different temperatures and storage times into different types [8]. This classification standard can be used with kinetic model and Arrhenius model to determine the quality index of Sichuan sauerkraut during storage. According to this standard, the shelf-life of Sichuan sauerkraut stored at different temperatures can be predicted and relatively accurate results can be obtained.

For the food that needs to be stored, the dynamic model can reduce the time cost in the storage process and improve the accuracy of predicting the quality index and shelf-life of the food [9]. Most foods contain a lot of nutrients and are very sensitive to temperature, so they are often in a state of high entropy and low enthalpy during storage [10]. The shelf-life model can represent the quality changes of food during storage. After Sichuan sauerkraut is produced, in order to determine the shelf-life of the product, it is usually determined by placing it at room temperature for a certain period of time. From a time-cost perspective, this is a time-consuming practice. Therefore, it is necessary to establish a shelf-life prediction model through accelerated experiments, which can reduce a lot of time costs. Reaction kinetics model was used to describe the quality changes of Sichuan sauerkraut, including hardness, color, total acid content, taste, and smell. The Arrhenius equation can be fitted according to the relationship between reaction rate constant k and temperature. Most foods are greatly affected by storage temperature, so Arrhenius equation is often used to describe the quality decay dynamics of such foods [11]. The main value of the Arrhenius relation is that data can be collected at high temperatures and then extrapolated to find shelf-life at lower temperatures. According to the experience of previous experiments, the accelerated experiments were carried out at 25 °C, 35 °C and 45 °C in this paper [12,13,14,15]. In addition, Gompertz model [16], Eyring model and Ball model [3] are often used to predict the shelf-life of foods. Through the combination of dynamic model and Arrhenius model, the relationship between quality index and time can be obtained.

## 2. Materials and Methods

### 2.1. Sichuan Sauerkraut Sample

Sichuan sauerkraut was provided by Sichuan Tianwei Food Co., Ltd. (Chengdu, China), which were vacuum-packed in polyethylene terephthalate bags with 250 g/bag. Radish and green vegetables are included in this Sichuan sauerkraut product. Sichuan sauerkraut were stored at different temperatures (25 °C, 35 °C and 45 °C) conditions with the humidity of 60% for 28 d, 14 d and 7 d, respectively.

### 2.2. Determination of Texture

The texture of Sichuan sauerkraut was determined by TA-XT2 texture analyzer (TA. XTC-18, Shanghai boshi Industrial Co., Ltd., Shanghai, China) equipped with a 4 mm cylindrical T/2 probe. The parameters are as follows: (1) Radish: the speed before the test is 5.0 mm/s, the speed is 2 mm/s, the speed is 2 mm/s after the test, the triggering force is 5.0 g, and the target value is 70%; (2) Vegetable: the speed before the test is 2.0 mm/s, the speed is 1 mm/s, the speed is 1 mm/s after the test, the triggering force is 5.0 g, and the target value is 40%. Each sample was determined ten times.

### 2.3. Determination of Color

The color of Sichuan sauerkraut was carried out by a colorimeter (Konica Minolta, CR-400, Osaka, Japan). The observation Angle of standard observer is about 2°, which conforms to CIE1931 standard. The light source *1 is C, D65. L*, a* and b* values are used to represent colors [17]. The more significant L* means the brighter the sample. a* values indicate red and green, and increases in a^*^ values indicate darker reds. b^*^ values are yellow and blue, larger b* correspond to dark blue [18]. The vegetables and radish were placed on the same background of the plate, and the samples were determined by the color meter, and each sample was determined ten times in parallel.

### 2.4. Determination of Total Acid

The total acid of Sichuan sauerkraut were measured according to the method of [19]. Deionized water was heated to a boiling state for 15 min to remove CO_2_, and then cooled to 70–80 °C. A 25 g sample was crushed in a beater. The sample was then mixed with deionized water, placed in conical flask with condenser, and boiled for 30 min. The sample was taken out, and after cooling to room temperature, the filtrate was collected by filtration with fast filter paper. Take 25 mL of test solution and put it in a beaker; put it on a magnetic stirrer, immerse the electrode of the acidity meter, and start the stirrer; quickly titrate with 0.1 mol/L NaOH solution to pH 8.20. When calculating the total acid content, use the conversion factor for lactic acid.

### 2.5. Microbiological Analysis

The total number of bacteria, lactic acid bacteria and mould in Sichuan sauerkraut were determined according to the methods provided in GB 4789.2-2016, GB 4789.35-2016 and GB 4789.15-2016 [20]. Where, the culture medium, culture temperature and time to determine the total number of bacteria are respectively plate counting agar medium (Sinopharm Chemical Reagent Co., Ltd., Shanghai, China), 37 °C and 48 h; to determine lactic acid bacteria are De Man, Rogosa and Sharp (MRS) agar medium (Hangzhou Best Biotechnology Co., Ltd., Hangzhou, China), 37 °C and 72 h; to determine mould are Bengal red agar medium (Sinopharm Chemical Reagent Co., Ltd.), 28 °C and 5 days. Data were represented by colony formation units (CFU) and results were expressed in CFU/g [21].

### 2.6. Determination of Flavor

The volatile components of Sichuan sauerkraut were analyzed by gas chromatography-mass spectrometry (GC-MS, Thermo Fisher Scientific, TSQ Quantum XLS, Waltham, MA, USA). Headspace solid phase extraction was used to extract volatile components from Sichuan sauerkraut. A 3.0 g sample was placed in a 15 mL gassed bottle. The setting procedure was: heat up to 40 °C for 3 min; At 3 °C/min to 160 °C; It then rose to 230 °C at 10 °C/min and held at 230 °C for 10 min. MS conditions: ion source temperature was 250 °C; ionization mode was EI; electron energy was 70 eV; mass scan range was 33~500 *m*/*z*.

### 2.7. Determination of Taste

E-tongue (Intelligent Sensor Technology Co., Kanagawa, Japan) was used to evaluate the taste properties of Sichuan sauerkraut, which consists of 8 taste sensors, two Ag/AgCl reference electrodes, an automatic sampler, a data acquisition system, and a data analysis system [22]. Each taste sensor has a special artificial lipid membrane, similar to the tongue [23].

### 2.8. Near Infrared Test

20 g of radish and green vegetable samples were taken for near-infrared (NIR) detection (IAS-3100, Wuxi Xunjie Guangyuan Technology Co., Ltd., Wuxi, China) each time. The NIR detector was calibrated by rotating the no-load detection disc. The start wavelength and end wavelength of this NIR detector were 900 nm and 1600 nm.

### 2.9. Sensory Evaluation

The samples of sensory evaluation were evaluated by 11 assessors who passed standardized sensory tests and evaluated the color, texture, smell, overall acceptance, saline taste and sour properties. The total sensory score of color, texture, smell, overall acceptance, saline taste and sour score was used to represent the overall sensory acceptance. The rating of each parameter was assigned separately using a 1–9 descriptive hedonic scale (9 = like extremely and 1 = dislike extremely). A sensory score of 1 was taken as the average score for minimum acceptability [24].

### 2.10. Dynamic Equation during Storage

During storage of Sichuan sauerkraut, the kinetic equation representing the change of quality index can be divided into zero-order, first-order and second-order kinetic reaction models. The kinetic models used in this experiment are zero-order and first-order kinetic reaction models, and their equations are:(1)$$\mathrm{Zero}\text{-}\mathrm{order}:\;A=A_{0}-kt$$
(2)$$\mathrm{First}\text{-}\mathrm{order}:\;A=A_{0}e^{-kt}$$
where, *A*: represent the value of quality index during storage; *A*_0_: initial value of quality index; *k*: reaction rate constant; *t*: storage time.

The value of *k* in the above equation is very dependent on temperature. In this study, Arrhenius model and Eyring model were used to represent the relationship between K value and temperature. The equation is as follows:(3)$$\mathrm{Arrhenius}\;\mathrm{equation}:\;\ln k=\ln k_{ref}-\frac{E_{a}}{R}\left(\frac{1}{T}-\frac{1}{T_{ref}}\right)$$
take the logarithm of both sides of this equation:(4)$$k=k_{ref}e^{-\frac{E_{a}}{R}\left(\frac{1}{T}-\frac{1}{T_{ref}}\right)}$$
where, $k_{ref}$: represents the frequency factor; $E_{a}$: represents the apparent activation energy (J/mol); *R*: is the gas constant, 8.3144 J/(mol·K); *T*: is the thermodynamic temperature during storage; *T_ref_* (K):is the reference temperature which was calculated using Equation (5).
(5)$$T_{ref}=\frac{1}{n}\sum_{i=1}^{n}T_{i}$$
where, *T_i_* is the studied temperature.
(6)$$\mathrm{Eyring}\;\mathrm{equation}:\;\ln\frac{k}{T}=-\frac{\Delta H^{*}}{R}\cdot \frac{1}{T}+\ln\frac{k_{B}}{h}+\frac{\Delta S^{*}}{R}$$

Take the logarithm of both sides of the equation and simplify it:(7)$$k=T\cdot e^{-\frac{\Delta H^{*}}{R}\cdot \frac{1}{T}+\ln\frac{k_{B}}{h}+\frac{\Delta S^{*}}{R}}$$
where, $\Delta H^{*}$ = enthalpy of activation; $\Delta S^{*}$ = entropy of activation; $k_{B}$ = Boltzmann constant, $1.381\times 10^{-23}\;$ J/K; *h* = Planck constant, $6.626\times 10^{-34\;}$ Js; *R* = gas constant, 8.3144 J/(mol·K); *T* = thermodynamic temperature during storage.

### 2.11. Establish BP-ANN Model

In many research fields, error back propagation artificial neural network (BP-ANN) is widely used to obtain more accurate prediction models. The characteristic of this method is its strong function approximation capability [25]. The BP-ANN model based on NIR parameters was constructed by using the input array of NIR spectral parameters and the output array of corresponding total acid content and storage days of 90 Sichuan sauerkraut samples [26]. Ninety Sichuan sauerkraut samples were divided into training set, validation set and prediction set, and cross-validation method was adopted. Each sample set accounted for 70%, 15% and 15% of the total number of samples, respectively, and the number of hidden layers of the model was 20 [27]. The network model fitting module of MATLAB R2020a (The Math Works Inc., Natick, MA, USA) was used to complete construction of BP-ANN model.

### 2.12. Data Analysis

All data were expressed as mean ± standard deviation by Microsoft Excel 2016. One-way analysis of variance (ANOVA) was applied in IBM SPSS 23.0 (IBM Inc., Armonk, NY, USA) to determine the significance level of differences between means, and 95% confidence level was used. Principal component analysis (PCA) is a method to transform related variables into unrelated variables, mainly using dimensionality reduction to compress multivariate data. Principal component analysis (PCA) and Fisher Discriminant analysis (FDA) were used in the taste response values of E-tongue to distinguish the tastes of samples at different storage periods. PCA and FDA data were processed by SPSS 23.0. All images used in this experiment are completed by Origin 2021. All the data obtained by the Arrhenius and Eyring models were analyzed by Origin 2021 and Microsoft Excel 2016.

## 3. Results and Discussion

### 3.1. Texture

During storage, the texture of Sichuan sauerkraut changed a lot, which was caused by the action of pectinase and cellulase. Cortes Rodriguez, et al. [28] found the same trend for texture. As can be seen from Table 1 in Sichuan sauerkraut stored at 45 °C, the hardness and chewiness of radish ranged from 763.28 to 148.97 gf, 93.04 to 35.67 gf, respectively. The hardness and chewiness of green vegetables ranged from 277.87 to 69.12 gf and 66.15 to 15.45 gf, respectively. At 35 °C, the hardness and chewiness of radish ranged from 763.28 to 113.25 gf, 93.04 to 17.88 gf, respectively. The hardness and chewiness of green vegetables ranged from 277.87 to 67.01 gf and 66.15 to 20.20 gf, respectively. At 25 °C, the hardness and chewiness of radish ranged from 763.28 to 182.95 gf, 93.04 to 25.52 gf, respectively. The hardness and chewiness of green vegetables ranged from 277.87 to 76.22 gf and 66.15 to 27.12 gf, respectively. The hardness and chewiness of Sichuan sauerkraut decreased by 85.16% and 86.56% respectively. As can be seen from Table 2, the k value of hardness increased with the increase of temperature (radish k value 1.7110 → 6.6641, green vegetables k value 0.5850 → 2.7775). The results indicated that temperature greatly influenced the hardness of radish and vegetables in Sichuan sauerkraut. The higher the temperature was, the decrease of hardness was faster.

### 3.2. Color

Color is the first indicator for consumers to judge the edibility of food, because the color of food can directly reflect the length of storage time. The color change of Sichuan sauerkraut is very obvious during storage, and the longer the storage time, the more obvious the color change. In this study, when stored at 45 °C, the L* value of radish in Sichuan pickled cabbage varies from 58.57 to 37.87, $a^{*}$ value varies from −3.06 to 3.19, b* value varies from 18.84 to 8.70, ΔE value varies from 18.77 to 8.83; L* value of vegetables was from 49.22 to 39.51, a* value was from −4.98 to −0.57, b* value was from 21.68 to 9.35, ΔE value from 4.86 to 16.44 (Table 3A). The Sichuan sauerkraut stored at 35 °C, L* value of radish in Sichuan sauerkraut varied from 58.57 to 42.13, a* value varied from −3.06 to 3.42, b* value varied from 18.84 to 13.01, ΔE value varied from 18.77 to 10.25; Vegetables showed L* values from 49.22 to 41.17, a* value was from −4.98 to −0.43, b* value was from 21.68 to 12.16, ΔE value from 4.86 to 13.36 (Table 3B). The Sichuan sauerkraut stored at 25 °C, L* value of radish in Sichuan sauerkraut varied from 58.57 to 44.79, a* value varied from −3.06 to 2.89, b* value varied from 18.84 to 14.47, ΔE value varied from 18.77 to 15.97; Vegetables showed L* values was from 49.22 to 44.98, a* value was from −4.98 to −1.39, b* value was from 21.68 to 15.22, ΔE value from 4.86 to 8.75 (Table 3C). During storage, the color of Sichuan sauerkraut will darken with time. This result is consistent with previous research [29,30]. The lower the temperature, the slower the color changes [31]. As can be seen from Table 2, L* value of radish and vegetables in Sichuan sauerkraut possesses a high degree of fit, which was suitable for the zero-order kinetic reaction equation.

### 3.3. Sensory Evaluation

Sensory evaluation is the evaluation of the quality of Sichuan sauerkraut in different storage times by consumers through eating, tasting, and providing scores with reference to certain standards. However, this method can more intuitively reflect the product in color, texture, flavor and other aspects of the acceptable degree. As can be seen from Figure 1, Sichuan sauerkraut is unacceptable at 25 °C, 35 °C and 45 °C when the storage time reached 336, 126 and 49 days, respectively. According to zero-order kinetic reaction model of sensory score, Sichuan sauerkraut had good fitting results at 25 °C (R^2^ 0.9895, RSME 0.2086), 35 °C (R^2^ 0.9713, RSME 0.2951) and 45 °C (R^2^ 0.9685, RSME 0.3111) (Table 2).

### 3.4. Total Acid

Sour taste is the main flavor sensation in Sichuan sauerkraut, and is also one of the characteristics of this food. Total acid is not only an important production index of Sichuan sauerkraut, but also an important parameter to judge its maturity or deterioration [32]. In contrast to hardness, *L** and sensory scores, total acid increases during storage [33] (Figure 2). The total acid content of Sichuan sauerkraut was found to range from 4.96 to 6.58 g/kg at 45 °C, whereas at 35 °C, the range is from 4.96 to 6.21 g/kg. During storage, total acid content can be increased by 36%, indicating that total acid is an important index affecting storage time of Sichuan sauerkraut. This is due to the fact that in an anaerobic environment lactic acid bacteria and some other microorganisms will consume sugars and increase the total acid content [34]. The results are in agreement with those determined by E-tongue.

### 3.5. Microorganism

As can be seen from Figure 3, the growth rate of microorganisms during storage is very slow and does not exceed the standard of food edible. As well, the reaction kinetics model is completely inconsistent with this index. This is due to the adding potassium sorbate and sodium D-isoascorbate during the processing of Sichuan sauerkraut after heat treatment. In this series of processing, greatly inhibited the growth of lactic acid bacteria and mold in Sichuan sauerkraut [1,35]. Although the number of microorganisms in Sichuan sauerkraut is very less, it has a certain impact on the quality of sauerkraut. With the increase of total acid content and high salt environment, the growth of microorganisms will be inhibited. Therefore, the microorganisms showed a trend of first rising and then falling during the storage process [36].

### 3.6. Volatiles and Taste

During storage, the change of flavor is also an important index affecting the edibility of Sichuan sauerkraut. GC-MS mainly measures the change of volatiles that affect smell, while e-tongue measures the change of taste. According to Table 4, a total of 27 flavor compounds were detected in Sichuan sauerkraut during storage. Among them, there are eight ester compounds, four aldehyde compounds, three thioether compounds, three acid compounds, three aromatic compounds, two sulfur compounds, two phenolic compounds, one ketone compound and one hydrocarbon compound. Among them, the main volatile substance in Sichuan sauerkraut includes Ethane, 1,1-bis(methylthio)-, Dimethyl trisulfide, Hexadecanoic acid, ethyl ester, Disulfide, dimethyl, Tetrasulfide, dimethyl, trans,trans-3,5-Heptadien-2-one, Sorbic Acid, Octanoic acid, Phenol, 4-ethyl- and Anethole. Ethers, acids, and aldehydes increase during storage whereas esters and aromatic compounds content decrease. Allyl Isothiocyanate is the special flavor component of Sichuan sauerkraut, which has certain anticancer activity. It gradually decreases during storage (45 °C: from 3.63 to 0.14; 35 °C from 3.63 to 0). This is due to the sensitivity of Allyl Isothiocyanate to temperature and pH [37,38]. This has a great impact on the flavor of Sichuan sauerkraut.

The analysis of e-tongue includes sourness, astringency, aftertaste-A, umami, richness and saltiness. It can be seen from Appendix A that the astringency, aftertaste-A, richness and saltiness of Sichuan sauerkraut did not change significantly during storage. This shows that these flavors are not the main reason for sauerkraut. The changes in sourness and umami were more obvious, which was consistent with the results of sensory evaluation. The sourness and umami of Sichuan sauerkraut were 14.79 to 17.18, 2.03 to 1.31 respectively when stored at 45 °C; at 35 °C, the sourness and umami were respectively 14.79 to 18.23, 2.03 to 1.40; at 25 °C, the sourness and umami were respectively 14.79 to 18.33, 2.03 to 1.43.

PCA adopts maximum variance method, and all relevant factors are analyzed. At 45 °C, PC1 and PC2 contributed 66.44% and 16.89%, respectively. The cumulative contribution of PC1 and PC2 is 83.33% (Figure 4a). At 35 °C, PC1 and PC2 contributed 64.66% and 16.87%, respectively. The cumulative contribution of PC1 and PC2 is 81.35% (Figure 4b). This indicates that these components could account for most of the data on flavor changes of Sichuan sauerkraut during storage [39].

PCA of electronic tongue can better classify the taste of Sichuan sauerkraut with different storage times [40]. As can be seen from the following Figure 4a,b, PCA of E-tongue index can be used to divide the storage period of Sichuan sauerkraut into three categories: completely acceptable period, acceptable period and unacceptable period. At 45 °C, fully acceptable period (0–28 days), acceptable period (28–56 days), unacceptable period (56–84 days); At 35 °C, fully acceptable period (0–42 days), acceptable period (42–126 days), unacceptable period (126–154 days) [7]. This was about the same as the sensory score.

FDA is another way to classify data based on variance and the classification and differentiation of the data. This method is used to verify the criteria of the above division [41]. Fisher function and non-standard function were selected for the analysis, while intra group correlation and group covariance were used for correlation of data. Generally, two-thirds of the data are extracted for training and one-third for testing. In the experiment, a total of 75 points were selected. There were 50 points as training data and 25 points as test data. The accuracy of these tests was more than 80%, indicating that the results were reliable.

Detailed classification is shown in Table 5 and Figure 5. For Sichuan sauerkraut that was stored at 45 °C, the Calibration data and Prediction data are 100% and 84.62%, respectively. The first classification type is fully acceptable period, calibration accuracy is 100%, prediction accuracy is 60%. The second classification type is the acceptable period, and the accuracy of calibration and prediction is 100%. The third classification type is unacceptable period. Calibration and Prediction accuracy are 100%. For Sichuan sauerkraut that was stored at 35 °C, the Calibration data and Prediction data are 100%.

The classification results of PCA and FDA were consistent for both Sichuan sauerkraut that were stored at 35 °C and 45 °C. This shows that these two classification results are credible.

### 3.7. NIR Spectrum Characteristics

Figure 6 shows the NIR spectra of Sichuan sauerkraut at different storage times. Obviously, there are three spectral absorption peaks in the NIR spectrum, which appear at 960 nm, 1150 nm and 1450 nm respectively. According to previous studies, the spectral characteristics in the band range of 960–980 nm and 1450 nm are related to water, while the absorption peak at 1170 nm is related to some organic compounds with C-H functional groups [42,43]. During storage, water content of Sichuan sauerkraut changes with storage time. With the increase of storage time, the intensity of the three spectral absorption peaks increased. This is due to the breakdown of the sauerkraut cell wall resulting in water loss [44].

### 3.8. Kinetics Model and Shelf-Life Prediction

Arrhenius and Eyring models were used to fit the relationship between k value in the zero-order reaction kinetics model based on the above quality indexes (Equations (5) and (7)) and the storage temperature. Since the k value of total acid is negative, absolute value of k value of total acid reaction kinetics model should be taken in modeling (Table 2). A series of parameters for successful fitting of Arrhenius and Eyring models are presented in Table 6. These two models can fit well the relationship between the change of k and temperature, except for the fitting of total acid in Eyring model.

In previous studies, the Arrhenius model has been widely applied, such as to aquatic products, fruits, vegetables, and so on [45,46,47]. With reference to the results obtained in this experiment, the Arrhenius model can well describe the changes of quality indexes of Sichuan sauerkraut during storage. The activation energy (*E_a_*) ranges from 47.23 to 72.09 kJ/mol, and the *k_ref_* varied from 1.076 × 10^6^–9.220 × 10^10^ d^−1^ (Table 2). Among them, the *E_a_* value of green vegetable color difference was the highest, followed by sensory evaluation, radish color difference, green vegetable hardness, radish hardness and total acid. The value of Δ*H** is close to the value of *E_a_*, which is in agreement with the findings presented by other researcher [9]. The value of Δ*H** and *E_a_* of total acid are the lowest, indicating that the increase of total acid does not need too high a temperature, and the higher the temperature, the more obvious the color change of Sichuan sauerkraut.

The activation enthalpy (Δ*H**) and entropy (Δ*S**) obtained from the Eyring model are the same as the above results. Different quality indexes have an influence on each parameter [48]. According to the above, the color has the highest $\Delta H^{*}$ value and *E_a_* value, indicating that it is the most sensitive to temperature, while the total acid is the least sensitive to temperature.

Compared with the Eyring model, the Arrhenius model exhibited the best fitting performance for each quality index (R^2^ value 0.9075–0.9995, RMSE value 0.0010–0.3766). Therefore, zero-order reaction kinetics model and Arrhenius model show better fitting effect, and they can better predict the shelf-life of Sichuan sauerkraut at different temperatures. According to Equations (1) and (4), Equation (8) can be used to predict the contents of quality indexes at different storage temperatures and storage times (Table 7).
(8)$$A=A_{0}-k_{ref}\times t\times e^{-\frac{E_{a}}{R}\left(\frac{1}{T}-\frac{1}{T_{ref}}\right)}$$

The method to determine the shelf-life of sauerkraut is to take the quality index of Sichuan sauerkraut as critical value. In this experiment, there was no specific critical value for radish color difference, vegetable color difference, radish hardness, vegetable hardness, and total acid, because the critical value of these quality indexes varied at different storage temperatures. However, the sensory score was set at a threshold of 1. According to sensory scores, the shelf-life of Sichuan sauerkraut at 25 °C, 35 °C, and 45 °C were 394, 158 and 67 days, respectively. According to the standard of e-tongue, the shelf-life of Sichuan sauerkraut can be defined as the maximum acceptable period of e-tongue. Therefore, the shelf-life of Sichuan sauerkraut at 25 °C, 35 °C and 45 °C are 365, 126 and 56 days respectively. At this point, at the end of the shelf-life of Sichuan sauerkraut, the color difference of radish, the color difference of vegetables, the hardness of radish, the hardness of vegetables and the total acid were close to 45.80, 41.69, 381.69 gf, 121.35 gf, 6.02 g/kg.

BP-ANN has the ability of self-learning and training, so it is used in this experiment to predict the shelf-life of Sichuan sauerkraut. In this study, 70 percent of the data was used for training, 15 percent for validation, and 15 percent for prediction. The modeling results are shown in Figure 7. The results show that the R^2^ of the training set, validation, set and prediction set are 0.9930, 0.9815, and 0.8731, respectively. The fitting degree of the modeling results is higher than that of the Arrhenius model based on the dynamics equation.

According to the above, it was predicted that the shelf-life storage time of Sichuan sauerkraut would decrease with the increase in temperature. Based on the combination of the zero-order dynamics equation with the Arrhenius model and BP-ANN model, the correlation between each index is quite high. Both models have different advantages. The zero-order kinetic equation combined with the Arrhenius model can better predict physical indexes such as texture and chromatic aberration, while the BP-ANN model can better predict chemical indexes such as total acid through NIR spectral parameters. The difference between different shelf-life predicted by radish color difference, vegetable color difference, radish hardness, vegetable hardness, and total acid was within 15%. Because different quality indexes changed differently at different temperatures, there were certain errors in predicting the shelf-life of Sichuan sauerkraut according to every single index. The results suggested that a single indicator was not accurate enough to predict the shelf-life, and a more accurate shelf-life can be predicted only when multiple indicators are combined.

## 4. Conclusions

In this paper, the changes in radish color difference, vegetable color difference, radish hardness, vegetable hardness, total acid, and sensory score of Sichuan Sauerkraut at storage temperatures 25, 35, and 45 °C were studied. The change of quality index of Sichuan sauerkraut during storage can be well fitted to the zero-order reaction kinetic model and BP-ANN model. The reaction rate k is obviously affected by temperature. According to the values of *E_a_* and Δ*H* obtained by Arrhenius and Eyring models, it can be seen that the color difference of vegetables in Sichuan sauerkraut is more sensitive to temperature than other quality indexes. The total acid content was the least affected by temperature and it increased with storage time. In general, in the long storage period, temperature plays a crucial role in the quality of sauerkraut, which will lead to the physical and chemical changes of Sichuan sauerkraut. As expected, Sichuan sauerkraut is better preserved at low temperatures.

This work studied the prediction of the shelf-life of Sichuan sauerkraut at different temperatures. The zero-order kinetic reaction model combined with the Arrhenius model could be used to predict the physical indexes of Sichuan sauerkraut, and the BP-ANN model could be used to predict the chemical indexes. The model can help dealers and consumers to better judge the storage time of the edible Sichuan sauerkraut. The E-tongue and GC-MS were used to determine the flavor and taste of Sichuan sauerkraut, and PCA and FDA were used to classify the e-tongue data. Three classification results were obtained, which were completely acceptable period, acceptable period, and unacceptable period respectively. By combining with zero-order kinetic reaction model and the Arrhenius model, the critical value of the Sichuan sauerkraut quality index was obtained, so as to determine the edible period of Sichuan sauerkraut more clearly.

It is feasible to store Sichuan sauerkraut by setting the temperature at 25 °C, 35 °C, and 45 °C through accelerated experiments. All kinds of data measured by this scheme can be well applied to Arrhenius and BP-ANN models. The reliability of the above model is proved by verifying the prediction accuracy of the model. Such models are of great significance in the production, transportation, and sales of Sichuan sauerkraut.

## Figures and Tables

**Figure 1 foods-11-01762-f001:**
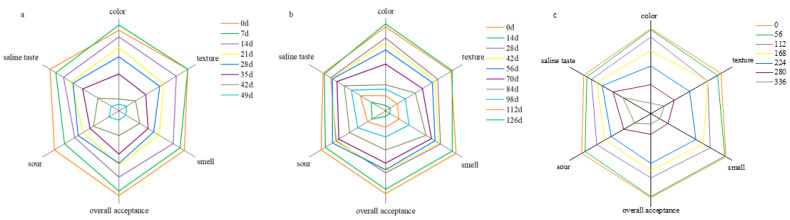
Sensory score obtained during storage at different times (the temperature of (**a**) is 45 °C; the temperature of (**b**) is 35 °C; the temperature of (**c**) is 25 °C).

**Figure 2 foods-11-01762-f002:**
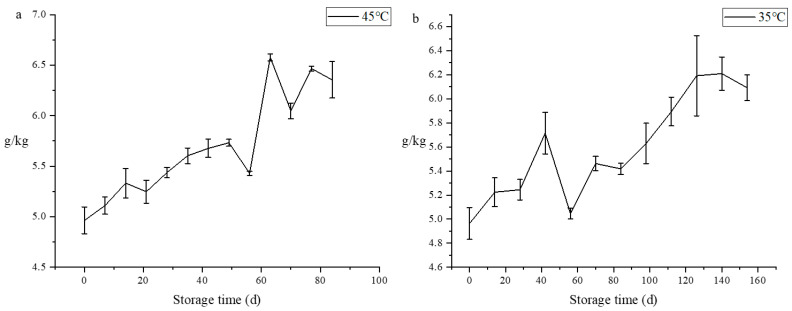
Changes of total acid in Sichuan sauerkraut during storage at different temperatures. (the temperature of (**a**) is 45 °C; the temperature of (**b**) is 35 °C.)

**Figure 3 foods-11-01762-f003:**
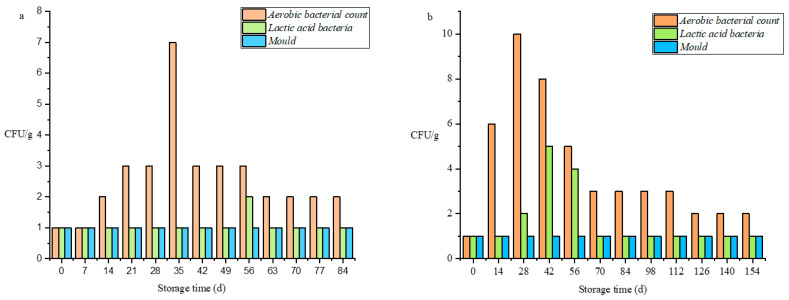
Changes of microorganism in Sichuan sauerkraut during storage at different temperatures (the temperature of (**a**) is 45 °C; the temperature of (**b**) is 35 °C).

**Figure 4 foods-11-01762-f004:**
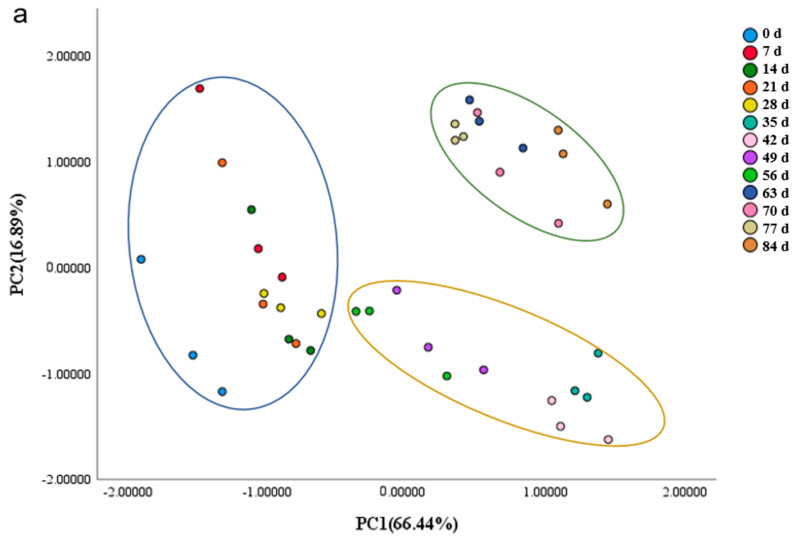
(**a**) PCA analysis of Sichuan sauerkraut during storage at 45 °C. (**b**) PCA analysis of Sichuan sauerkraut during storage at 35 °C.

**Figure 5 foods-11-01762-f005:**
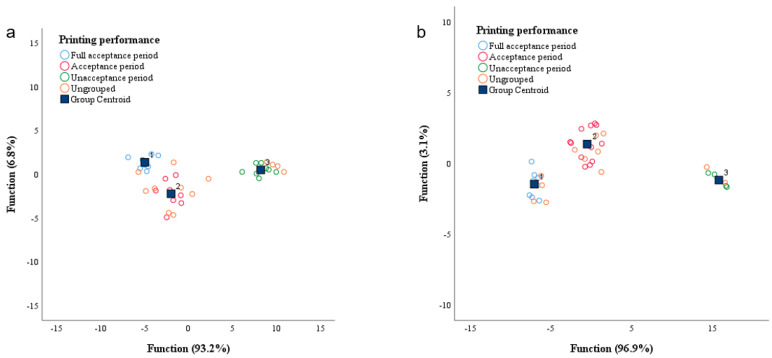
Fisher discriminant analysis plot of Sichuan sauerkraut during storage at different temperatures (the temperature of (**a**) is 45 °C; the temperature of (**b**) is 35 °C; 1: Full acceptance period; 2: Acceptance period; 3: Unacceptance period).

**Figure 6 foods-11-01762-f006:**
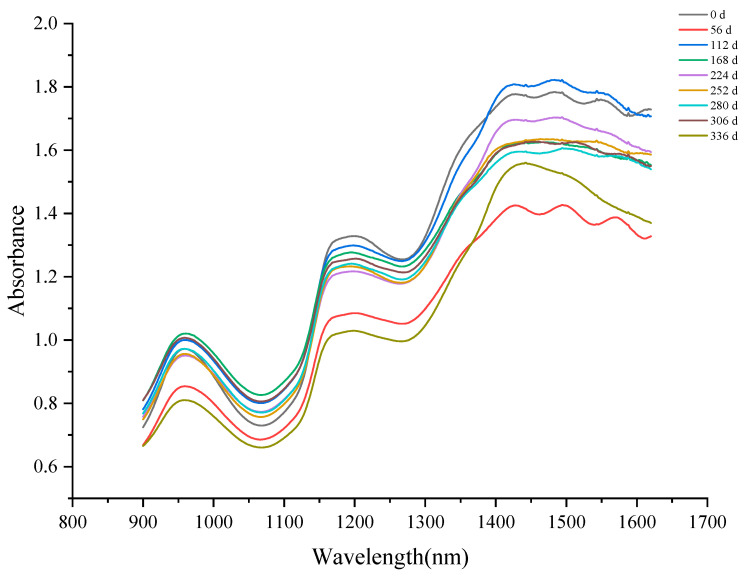
NIR analysis of Sichuan sauerkraut during storage.

**Figure 7 foods-11-01762-f007:**
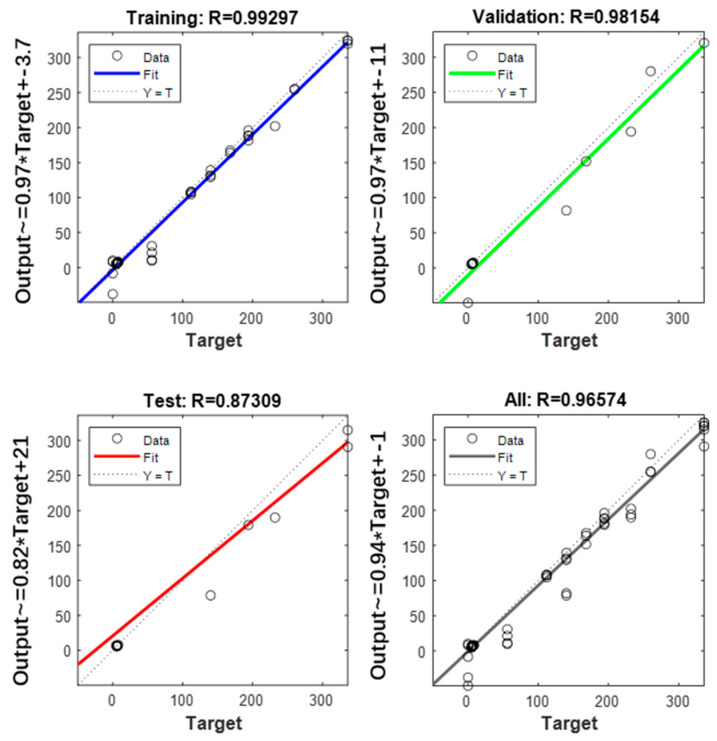
BP-ANN model fitting results for total acid content and storage days of Sichuan Sauerkraut based on NIR.

**Table 1 foods-11-01762-t001:** Texture changes of Sichuan sauerkraut at different storage temperatures.

Storage Time (d)	Temperature (45 °C)			
	Hardness (gf)/Rd	Chewiness (gf)/Rd	Hardness (gf)/V	Chewiness (gf)/V
0	763.28 ± 57.54 ^a^	93.04 ± 49.48 ^a^	277.87 ± 78.42 ^a^	66.15 ± 21.45 ^a^
7	634.29 ± 141.37 ^b^	54.30 ± 17.49 ^b^	194.18 ± 71.49 ^bc^	59.19 ± 19.93 ^a^
14	760.48 ± 98.12 ^a^	34.16 ± 12.64 ^b^	228.13 ± 79.50 ^b^	54.17 ± 19.57 ^ab^
21	605.33 ± 121.40 ^b^	46.19 ± 19.76 ^b^	200.42 ± 77.80 ^bc^	43.02 ± 18.63 ^bc^
28	458.82 ± 93.37 ^c^	40.60 ± 22.33 ^b^	152.87 ± 26.93 ^c^	41.82 ± 16.66 ^bc^
35	344.55 ± 105.05 ^d^	39.89 ± 24.69 ^b^	69.90 ± 25.88 ^d^	28.17 ± 10.95 ^cde^
42	251.28 ± 72.92 ^de^	35.73 ± 10.67 ^b^	98.89 ± 38.48 ^d^	34.92 ± 13.64 ^cd^
49	233.73 ± 71.30 ^de^	46.84 ± 14.65 ^b^	68.27 ± 30.23 ^d^	23.58 ± 12.61 ^de^
56	191.81 ± 72.85 ^e^	41.83 ± 26.04 ^b^	79.13 ± 35.95 ^d^	28.39 ± 13.87 ^cde^
63	249.18 ± 109.63 ^de^	41.35 ± 32.65 ^b^	65.08 ± 29.63 ^d^	20.48 ± 8.30 ^de^
70	243.71 ± 32.53 ^de^	29.36 ± 10.84 ^b^	64.18 ± 22.86 ^d^	18.99 ± 9.25 ^de^
77	219.84 ± 55.44 ^e^	27.63 ± 8.75 ^b^	58.17 ± 15.18 ^d^	20.47 ± 5.45 ^de^
84	148.97 ± 29.84 ^e^	35.67 ± 12.72 ^b^	69.12 ± 18.84 ^d^	15.45 ± 7.84 ^e^
Storage time (d)	Temperature (35 °C)			
0	763.28 ± 57.54 ^a^	93.04 ± 49.48 ^a^	277.87 ± 78.42 ^a^	66.15 ± 21.45 ^a^
14	691.83 ± 114.67 ^ab^	70.49 ± 33.83 ^abc^	259.48 ± 16.05 ^a^	82.72 ± 25.60 ^a^
28	635.23 ± 117.83 ^b^	60.41 ± 25.96 ^abcd^	192.59 ± 49.61 ^b^	52.98 ± 15.82 ^bc^
42	644.19 ± 107.13 ^b^	79.25 ± 44.10 ^ab^	177.50 ± 53.40 ^b^	48.95 ± 16.51 ^c^
56	596.58 ± 117.58 ^b^	49.30 ± 24.16 ^bcde^	165.28 ± 45.77 ^b^	49.94 ± 13.54 ^c^
70	420.33 ± 94.18 ^c^	58.08 ± 27.58 ^abcd^	108.13 ± 61.94 ^cd^	33.63 ± 14.52 ^de^
84	181.78 ± 73.94 ^de^	64.52 ± 25.51 ^abc^	80.38 ± 27.95 ^cde^	30.45 ± 10.70 ^def^
98	261.48 ± 76.66 ^d^	40.08 ± 17.16 ^cdef^	86.08 ± 29.92 ^cde^	26.07 ± 8.30 ^def^
112	251.32 ± 67.48 ^d^	35.66 ± 15.44 ^cdef^	109.68 ± 23.22 ^c^	35.67 ± 9.12 ^d^
126	240.09 ± 71.86 ^d^	25.21 ± 27.68 ^def^	59.91 ± 19.58 ^e^	18.73 ± 6.77 ^f^
140	94.92 ± 24.35 ^e^	12.50 ± 7.24 ^f^	64.28 ± 22.22 ^e^	18.37 ± 7.76 ^f^
154	113.25 ± 30.51 ^e^	17.88 ± 7.14 ^ef^	67.01 ± 15.92 ^de^	20.20 ± 8.01 ^ef^
Storage time (d)	Temperature (25 °C)			
0	763.28 ± 57.54 ^a^	93.04 ± 49.48 ^a^	277.87 ± 78.42 ^a^	66.15 ± 21.45 ^a^
56	730.98 ± 108.74 ^a^	67.41 ± 23.88 ^b^	273.82 ± 29.94 ^a^	58.75 ± 14.36 ^ab^
112	659.29 ± 76.18 ^a^	56.93 ± 14.59 ^bc^	189.99 ± 40.19 ^b^	65.87 ± 18.52 ^a^
168	403.41 ± 100.74 ^b^	57.12 ± 15.59 ^bc^	216.67 ± 45.01 ^b^	46.69 ± 15.85 ^abc^
224	408.97 ± 106.85 ^b^	45.81 ± 24.20 ^cd^	221.66 ± 59.50 ^b^	44.39 ± 15.51 ^bc^
280	268.74 ± 61.49 ^c^	35.38 ± 15.85 ^de^	135.88 ± 37.52 ^c^	40.22 ± 19.91 ^bc^
336	182.95 ± 63.06 ^c^	25.52 ± 10.21 ^e^	76.22 ± 32.39 ^d^	27.12 ± 11.08 ^c^

Rd = Radish; V = Vegetable. Different letters following the average ± deviation in a column mean significant differences (*p* < 0.05).

**Table 2 foods-11-01762-t002:** Reaction order estimation of quality indices based on the coefficient of determination (R^2^) and root mean square error (RMSE) from the zero- and first-order.

Quality Indices	T (°C)	Zero-Order			First-Order		
		k	R^2^	RSME	k	R^2^	RSME
Total acid	25	−0.0078	0.8699	0.3384	--	--	--
	35	−0.0083	0.9523	0.0958	−0.0015	0.9494	0.0968
	45	−0.0196	0.9684	0.0943	−0.0034	0.9738	0.0888
L* _Rd_	25	0.0424	0.9325	1.3799	--	--	--
	35	0.1035	0.9424	1.2946	0.0022	0.9657	1.1442
	45	0.2276	0.9029	2.0287	0.0050	0.9468	1.7736
L* _v_	25	0.0238	0.8906	0.9210	--	--	--
	35	0.0536	0.9274	0.7635	0.0012	0.9202	0.7807
	45	0.1346	0.9188	1.0812	0.0030	0.9265	1.0507
Hardness _Rd_	25	1.7110	0.9694	34.0525	--	--	--
	35	4.3592	0.9669	40.4722	0.0097	0.8781	54.3007
	45	6.6641	0.8761	70.3778	0.0019	0.9596	49.5786
Hardness _V_	25	0.5850	0.9840	8.8842	--	--	--
	35	1.4207	0.9063	23.5509	0.0114	0.9967	11.6397
	45	2.7775	0.9116	23.0117	0.0219	0.9749	11.5778
Sensory score	25	0.0212	0.9895	0.2086	--	--	--
	35	0.0427	0.9713	0.2951	0.0079	0.9243	0.3661
	45	0.1075	0.9685	0.3111	0.0180	0.8772	0.4704

--: No detection; _Rd_ = radish; _V_ = vegetable; T = temperature; k = reaction rate.

**Table 3 foods-11-01762-t003:** (**A**) Color change of Sichuan sauerkraut during storage at 45 °C. (**B**) Color change of Sichuan sauerkraut during storage at 35 °C. (**C**) Color change of Sichuan sauerkraut during storage at 25 °C.

(**A**)				
Storage Time (d)	Temperature (45 °C)			
	Sample (radish)			
	L*	a*	b*	∆E
0	58.57 ± 2.60 ^c^	−3.06 ± 0.32 ^g^	18.84 ± 0.49 ^a^	0.00 ± 0.00 ^g^
7	56.88 ± 2.20 ^a^	0.54 ± 0.30 ^f^	17.31 ± 2.00 ^ab^	5.35 ± 1.95 ^f^
14	53.20 ± 2.29 ^b^	1.39 ± 0.43 ^e^	18.28 ± 2.40 ^a^	7.38 ± 1.65 ^f^
21	52.36 ± 2.49 ^b^	1.58 ± 0.31 ^f^	17.39 ± 1.58 ^ab^	7.31 ± 2.25 ^f^
28	45.66 ± 4.08 ^cde^	2.62 ± 0.42 ^cd^	16.19 ± 2.29 ^abc^	14.61 ± 3.68 ^de^
35	45.61 ± 1.99 ^cde^	2.36 ± 0.62 ^d^	14.38 ± 2.02 ^cd^	14.87 ± 2.00 ^de^
42	43.73 ± 4.03 ^def^	3.05 ± 0.96 ^bcd^	12.28 ± 3.18 ^d^	17.64 ± 3.94 ^cde^
49	46.05 ± 3.98 ^cd^	2.51 ± 0.72 ^d^	14.74 ± 1.57 ^bcd^	14.46 ± 3.68 ^e^
56	42.97 ± 5.25 ^def^	2.96 ± 0.77 ^cd^	13.26 ± 3.78 ^d^	17.96 ± 5.39 ^cd^
63	42.52 ± 2.88 ^ef^	3.87 ± 1.42 ^ab^	13.09 ± 3.03 ^d^	18.69 ± 2.82 ^bc^
70	41.00 ± 2.32 ^fg^	4.42 ± 1.38 ^a^	13.74 ± 3.59 ^cd^	20.08 ± 2.50 ^bc^
77	40.57 ± 2.65 ^fg^	3.46 ± 1.06 ^bc^	9.60 ± 2.38 ^e^	21.39 ± 2.73 ^ab^
84	37.87 ± 2.46 ^g^	3.19 ± 0.83 ^bcd^	8.70 ± 1.77 ^fd^	23.94 ± 2.58 ^a^
Storage time (d)	Sample (vegetable)			
0	49.22 ± 3.01 ^a^	−4.98 ± 0.83 ^f^	21.68 ± 2.54 ^a^	0.00 ± 0.00 ^f^
7	49.70 ± 3.65 ^a^	−2.73 ± 0.96 ^e^	21.89 ± 3.11 ^a^	4.86 ± 1.90 ^e^
14	51.15 ± 3.76 ^a^	−3.20 ± 0.78 ^e^	22.78 ± 3.83 ^a^	5.51 ± 2.17 ^e^
21	49.78 ± 1.99 ^a^	−2.87 ± 0.72 ^e^	21.02 ± 2.82 ^a^	3.61 ± 1.93 ^e^
28	44.38 ± 2.34 ^b^	−1.21 ± 0.67 ^bcd^	15.91 ± 2.59 ^b^	8.85 ± 2.08 ^d^
35	42.60 ± 2.23 ^bd^	−1.47 ± 1.00 ^cd^	14.14 ± 3.23 ^bc^	10.87 ± 3.24 ^cd^
42	44.64 ± 1.85 ^b^	−1.32 ± 0.45 ^cd^	14.32 ± 2.51 ^bc^	9.61 ± 2.40 ^cd^
49	43.03 ± 3.60 ^bc^	−1.49 ± 0.67 ^d^	14.28 ± 3.36 ^bc^	10.54 ± 4.26 ^cd^
56	42.78 ± 3.09 ^bd^	−0.37 ± 1.35 ^ab^	12.88 ± 2.45 ^cd^	12.19 ± 2.86 ^bc^
63	41.16 ± 2.15 ^cd^	−0.04 ± 1.03 ^a^	11.58 ± 2.40 ^cde^	14.01 ± 2.47 ^ab^
70	39.64 ± 3.28 ^d^	−0.71 ± 1.06 ^abcd^	12.30 ± 2.16 ^cd^	14.26 ± 3.24 ^ab^
77	40.61 ± 2.16 ^cd^	0.15 ± 0.92 ^a^	10.15 ± 2.28 ^de^	15.38 ± 2.68 ^a^
84	39.51 ± 2.56 ^d^	−0.57 ± 1.08 ^abc^	9.35 ± 3.31 ^e^	16.44 ± 3.67 ^a^
(**B**)				
Storage time (d)	Temperature (35 °C)			
	Sample (radish)			
	L*	a*	b*	∆E
0	58.57 ± 2.60 ^b^	−3.06 ± 0.32 ^f^	18.84 ± 0.49 ^abc^	0.00 ± 0.00 ^d^
14	54.22 ± 4.02 ^a^	−0.13 ± 0.81 ^e^	15.28 ± 1.04 ^bcd^	6.86 ± 3.21 ^c^
28	53.15 ± 2.17 ^ab^	0.67 ± 0.5 ^cd^	16.47 ± 2.23 ^abc^	7.41 ± 1.84 ^c^
42	52.00 ± 3.11 ^ab^	0.45 ± 0.54 ^de^	17.36 ± 2.55 ^ab^	8.16 ± 2.59 ^bc^
56	50.56 ± 4.85 ^b^	1.32 ± 0.70 ^c^	17.32 ± 3.12 ^ab^	9.84 ± 4.62 ^bc^
70	49.70 ± 4.22 ^b^	3.10 ± 0.77 ^a^	17.94 ± 3.07 ^a^	11.43 ± 3.65 ^b^
84	41.76 ± 3.23 ^b^	1.30 ± 0.86 ^c^	13.41 ± 2.97 ^d^	18.39 ± 3.46 ^a^
98	44.89 ± 2.35 ^d^	2.22 ± 0.80 ^b^	14.36 ± 3.16 ^cd^	15.58 ± 2.75 ^a^
112	42.59 ± 4.10 ^cd^	2.92 ± 0.86 ^ab^	13.89 ± 3.11 ^cd^	18.02 ± 4.10 ^a^
126	46.07 ± 3.30 ^cd^	2.92 ± 0.55 ^ab^	12.72 ± 2.44 ^e^	15.58 ± 3.30 ^a^
140	42.84 ± 2.00 ^c^	3.58 ± 1.24 ^a^	14.04 ± 2.74 ^cd^	17.95 ± 2.09 ^a^
154	42.13 ± 2.62 ^d^	3.42 ± 0.80 ^a^	13.01 ± 2.75 ^d^	18.81 ± 2.54 ^a^
	Sample (vegetable)			
0	49.22 ± 3.01 ^b^	−4.98 ± 0.83 ^e^	21.68 ± 2.54 ^ab^	0.00 ± 0.00 ^e^
14	48.24 ± 3.67 ^bc^	−2.52 ± 1.25 ^cd^	23.06 ± 2.25 ^a^	4.94 ± 1.69 ^d^
28	49.19 ± 2.77 ^b^	−2.46 ± 0.94 ^d^	22.48 ± 4.69 ^a^	5.59 ± 1.90 ^d^
42	52.05 ± 2.79 ^a^	−3.19 ± 0.71 ^d^	24.00 ± 3.41 ^a^	5.20 ± 2.89 ^d^
56	48.69 ± 2.75 ^bc^	−1.78 ± 1.09 ^bc^	22.75 ± 3.40 ^a^	5.07 ± 2.17 ^d^
70	46.24 ± 1.92 ^cd^	−1.12 ± 1.06 ^ab^	19.47 ± 4.19 ^b^	6.41 ± 2.91 ^d^
84	45.24 ± 3.68 ^de^	−2.39 ± 0.62 ^cd^	16.52 ± 1.90 ^c^	7.52 ± 3.05 ^cd^
98	44.10 ± 1.93 ^de^	−1.41 ± 0.92 ^b^	14.86 ± 3.02 ^cd^	9.40 ± 3.25 ^bc^
112	42.66 ± 2.71 ^ef^	−1.00 ± 0.65 ^ab^	14.93 ± 2.62 ^cd^	10.40 ± 3.24 ^b^
126	44.06 ± 1.94 ^de^	−0.85 ± 0.70 ^ab^	13.43 ± 1.72 ^d^	10.71 ± 1.94 ^b^
140	43.04 ± 2.86 ^ef^	−0.26 ± 1.46 ^a^	14.38 ± 2.74 ^cd^	10.99 ± 3.14 ^b^
154	41.17 ± 1.51 ^f^	−0.43 ± 1.11 ^a^	12.16 ± 2.98 ^d^	13.36 ± 3.07 ^a^
(**C**)				
Storage time (d)	Temperature (25 °C)			
	Sample (radish)			
	L*	a*	b*	∆E
0	58.57 ± 2.60 ^a^	−3.06 ± 0.32 ^e^	18.84 ± 0.49 ^ab^	0.00 ± 0.00 ^d^
56	51.78 ± 3.36 ^a^	1.12 ± 0.53 ^c^	17.91 ± 1.46 ^ab^	8.48 ± 2.30 ^c^
112	50.44 ± 2.81 ^ab^	0.84 ± 0.41 ^c^	17.97 ± 2.01 ^ab^	9.32 ± 2.59 ^c^
168	52.13 ± 2.15 ^a^	0.22 ± 0.37 ^d^	18.79 ± 2.17 ^a^	7.61 ± 1.79 ^c^
224	46.30 ± 2.83 ^cd^	2.28 ± 0.48 ^b^	16.05 ± 2.24 ^bc^	13.85 ± 2.79 ^b^
280	42.45 ± 2.47 ^e^	2.29 ± 0.59 ^b^	12.74 ± 2.56 ^d^	18.20 ± 2.64 ^a^
336	44.79 ± 3.73 ^de^	2.89 ± 0.83 ^a^	14.47 ± 3.55 ^cd^	15.97 ± 3.96 ^ab^
	Sample (vegetable)			
0	49.22 ± 3.01 ^a^	−4.98 ± 0.83 ^d^	21.68 ± 2.54 ^a^	0.00 ± 0.00 ^d^
56	48.57 ± 3.24 ^a^	−2.07 ± 0.65 ^bc^	21.33 ± 2.27 ^ab^	4.65 ± 1.42 ^c^
112	49.99 ± 2.56 ^a^	−1.78 ± 0.97 ^abc^	22.94 ± 3.14 ^a^	5.08 ± 1.56 ^c^
168	48.54 ± 2.72 ^a^	−2.52 ± 0.69 ^c^	22.21 ± 3.96 ^a^	4.82 ± 2.31 ^c^
224	45.57 ± 1.27 ^b^	−1.13 ± 0.77 ^a^	18.33 ± 2.86 ^bc^	6.64 ± 2.26 ^bc^
280	42.67 ± 1.58 ^c^	−1.43 ± 0.35 ^ab^	15.62 ± 1.83 ^c^	9.74 ± 1.71 ^a^
336	44.98 ± 2.30 ^bc^	−1.39 ± 0.61 ^ab^	15.22 ± 2.12 ^c^	8.75 ± 2.38 ^ab^

Different letters following the average ± deviation in a column mean significant differences (*p* < 0.05).

**Table 4 foods-11-01762-t004:** Changes of volatile components in Sichuan sauerkraut during storage (45 °C, 35 °C).

Classification	Molecular Formula	Compounds	RT		Storage Temperature					
					45 °C			35 °C		
				0 d	28 d	56 d	84 d	42 d	70 d	154 d
				Area %	Area %	Area %	Area %	Area %	Area %	Area %
Sulfur compounds (2)										
1	C_4_H_10_S_2_	Ethane, 1,1-bis (methylthio) -	9.12	8.97	6.76	1.79	0.07	9.85	8.23	1.9
2	C_2_H_6_S_3_	Dimethyl trisulfide	10.36	4.62	4.85	8.63	8.59	5.94	6.8	4.02
Ester compounds (8)										
1	C_5_H_10_O_3_	Propanoic acid, 2-hydroxy-, ethyl ester	4.82	1.28	1.51	1	0.86	1.25	1.36	2.63
2	C_10_H_20_O_2_	Octanoic acid, ethyl ester	21.24	0.39	1.04	0.24	0.26	1.23	1.43	0.75
3	C_17_H_34_O_2_	Hexadecanoic acid, methyl ester	46.88	0.66	0.37	0.41	0.06	0.32	0.27	0.09
4	C_18_H_36_O_2_	Hexadecanoic acid, ethyl ester	47.84	0.5	1.96	1.23	0.02	1.17	0.92	0.15
5	C_4_H_5_NS	Allyl Isothiocyanate	6.61	3.63	0.44	0.37	0.14	0.98	0.44	--
6	C_7_H_14_O_2_	1-Butanol, 3-methyl-, acetate	6.86	0.13	0.12	0.08	0.48	0.19	0.15	0.12
7	C_12_H_24_O_2_	Decanoic acid, ethyl ester	29.88	0.2	0.29	0.69	--	0.14	0.18	0.06
8	C_14_H_28_O_2_	Dodecanoic acid, ethyl ester	44.49	0.02	2.23	0.05	--	0.09	0.07	--
Thioethers compounds (3)										
1	C_2_H_6_S_2_	Disulfide, dimethyl	3.18	9.78	6.5	9.48	8.94	9.04	7.18	4.19
2	C_2_H_6_S_4_	Dimethyl tetrasulfide	21.61	0.34	1.16	4.17	2.99	0.55	0.63	4.19
3	C_6_H_10_S_3_	Trisulfide, di-2-propenyl	25.51	0.32	0.16	--	--	--	0.16	--
Ketone compounds (1)										
1	C_7_H_10_O	trans, trans-3,5-Heptadien-2-one	11.65	0.35	10.13	11.74	10.91	1.06	1.43	3.99
Acid compounds (3)										
1	C_6_H_8_O_2_	Sorbic Acid	16.49	10.75	13.59	14.23	3.36	18.69	13.28	11.02
2	C_8_H_16_O_2_	Octanoic acid	20.79	--	--	0.88	1.06	--	--	0.14
3	C_10_H_20_O_2_	n-Decanoic acid	20.95	--	--	3.27	0.32	0.29	0.22	0.75
Phenolic compounds (2)										
1	C_8_H_10_O	Phenol, 4-ethyl-	20.3	3.48	6.03	3.36	1.38	5.08	5.91	5.06
2	C_9_H_12_O_2_	Phenol, 4-ethyl-2-methoxy-	24.82	1.3	2.19	1.09	0.34	2.05	2.11	1.32
Aromatic compounds (3)										
1	C_10_H_12_O	Anethole	25.08	7.38	1.09	--	--	--	1.09	--
2	C_9_H_9_N	Benzenepropanenitrile	22.99	--	0.51	0.44	0.12	0.58	0.6	0.4
3	C_10_H_10_O	1H-Indene-4-carboxaldehyde, 2,3-dihydro- (9CI)	24.42	--	1.69	1.2	0.51	--	0.78	0.98
Aldehyde compounds (4)										
1	C_5_H_4_O_2_	Furfural	5.38	--	1.04	0.86	0.98	0.55	0.56	1.87
2	C_7_H_6_O	Benzaldehyde	10.1	0.17	0.66	1.23	2.33	0.66	0.66	2.05
3	C_8_H_8_O_2_	Benzaldehyde, 4-methoxy-	23.61	1.24	0.21	--	--	--	0.21	--
4	C_7_H_10_O	2,4-Heptadienal, (E, E)-	12.42	--	--	0.61	1.48	--	--	--
Hydrocarbon compounds (1)										
1	C_8_H_14_	5,5-Dimethyl-1,3-hexadiene-	13.75	--	2.49	2.4	0.02	--	2.82	0.11

--: No detection; RT = appearance time; Area = peak area.

**Table 5 foods-11-01762-t005:** (**A**) Fisher discriminant analysis was used to classify Sichuan sauerkraut during storage at 45 °C. (**B**) Fisher discriminant analysis was used to classify Sichuan sauerkraut during storage at 35 °C.

(**A**)										
Sample-45 °C (storage time/d)	Calibration data					Prediction data				
	Full acceptance period	Acceptance period	Unacceptance period	Total	Correctly classified (%)	Full acceptance period	Acceptance period	Unacceptance period	Total	Correctly classified (%)
0, 7, 14, 21, 28	10			10	100	3	2		5	60
35, 42, 49, 56		8		8	100		4		4	100
63, 70, 77, 84			8	8	100			4	4	100
Total				26	100				13	84.62
(**B**)										
Sample-35 °C (storage time/d)	Calibration data					Prediction data				
	Full acceptance period	Acceptance period	Unacceptance period	Total	Correctly classified (%)	Full acceptance period	Acceptance period	Unacceptance period	Total	Correctly classified (%)
0, 14, 28, 42	8			8	100	4			4	100
56, 70, 84, 98, 112, 126		12		12	100		6		6	100
140, 154			4	4	100			2	2	100
Total				24	100				12	100

**Table 6 foods-11-01762-t006:** Kinetic parameters of quality indices during storage obtained by the Arrhenius and Eyring models.

Quality Indices	Arrhenius Model				Eyring Model			
	$E_{a}\;(\mathrm{kJ}/\mathrm{mol})$	$k_{ref}\;(d^{-1})$	R^2^	RSME	∆H* (kJ/mol)	∆S* (J/(mol·K))	R^2^	RSME
Total acid	47.23	−1.0616	0.9567	0.0018	--	--	--	--
L* _Rd_	65.18	0.1024	0.9943	0.0010	62.60	−60.98	0.9942	0.0010
L* _v_	72.09	0.0554	0.9995	0.0017	69.51	−43.66	0.9995	0.0017
Hardness _Rd_	45.76	3.8813	0.9075	0.3766	43.19	−93.75	0.9075	0.3774
Hardness _v_	58.10	1.3682	0.9779	0.0443	55.52	−62.38	0.9778	0.0446
Sensory score	70.07	0.0453	0.9955	0.0024	67.48	−51.92	0.9995	0.0023

--: No detection; _Rd_ = radish; _V_ = vegetable; $E_{a}$ = apparent activation energy (kJ/mol); $k_{ref}$ = reaction rate at the reference temperature (d^−1^); ∆H* = enthalpy of activation (kJ/mol); ∆S* = entropy of activation; R^2^ = coefficient of determination; RSME = root mean square error.

**Table 7 foods-11-01762-t007:** Arrhenius model for predicting various quality indexes of Sichuan sauerkraut in different time.

Quality Indexes	Prediction Model
Total acid	$t=\frac{\left|4.96-A\right|}{1.076\times 10^{6}}\times e^{\frac{47.23\times 10^{3}}{R}\left(\frac{1}{T}-\frac{1}{T_{ref}}\right)}$
L* _Rd_	$t=\frac{\left(58.57-B\right)}{1.147\times 10^{10}}\times e^{\frac{65.18\times 10^{3}}{R}\left(\frac{1}{T}-\frac{1}{T_{ref}}\right)}$
L* _v_	$t=\frac{\left(49.22-C\right)}{9.220\times 10^{10}}\times e^{\frac{72.09\times 10^{3}}{R}\left(\frac{1}{T}-\frac{1}{T_{ref}}\right)}$
Hardness _Rd_	$t=\frac{\left(763.28-D\right)}{2.221\times 10^{8}}\times e^{\frac{45.76\times 10^{3}}{R}\left(\frac{1}{T}-\frac{1}{T_{ref}}\right)}$
Hardness _v_	$t=\frac{\left(277.87-E\right)}{9.673\times 10^{9}}\times e^{\frac{58.10\times 10^{3}}{R}\left(\frac{1}{T}-\frac{1}{T_{ref}}\right)}$
Sensory score	$t=\frac{\left(8.11-F\right)}{3.412\times 10^{10}}\times e^{\frac{70.07\times 10^{3}}{R}\left(\frac{1}{T}-\frac{1}{T_{ref}}\right)}$

_Rd_ = radish; _V_ = vegetable; *R* = gas constant, 8.3144 J/(mol·K); *T* = thermodynamic temperature during storage. *A*, *B*, *C*, *D*, *E*, *F*: represents the measured value of the corresponding quality indexes; *T_ref_* = the studied temperature.

## Data Availability

The data presented in this study are available on request from the corresponding author.

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
