# Peer review of "Shelf-Life Prediction and Critical Value of Quality Index of Sichuan Sauerkraut Based on Kinetic Model and Principal Component Analysis"

_foods, 2022, doi:10.3390/foods11121762_

Round 1
Reviewer 1 Report
REVIEW REPORT FOODS 2022
ABSTRACT section
Lines 21-23: „The alteration of texture, color,...“ is an unclear sentence; it should be reformulated.
Line 22: replace „flavor“ with „volatile components“
Lines 24-27: „Principal component analysis...“ is also unclear sentence; it should be reformulated.
Lines 35-37: „The establishment of the model...“ is also unclear sentence; it should be reformulated.
INTRODUCTION section
Lines 48-58: These sentences are not clear enough and need to be reformulated.
Lines 59-63: These sentences are not clear enough and need to be reformulated.
In general, the Introduction section should be revised in terms of language. Also, it is not clear why the authors have chosen these high storage temperatures. Are these storage temeperatures of Sichuan sauerkraut common in practice, or in any part of the supply chain? This should be described more precisely.
MATERIALS AND METHODS section
Line 97: Radish (??????). It should be better explained (what is the meaning of terms „radish“ and „vegetables“ in texture analysis). According to the title, the research is aimed at studying the change in the quality of sauerkraut, not radish or vegetables.
Line 99: Vegetable (??????). It should be better explained (what is the meaning of terms „radish“ and „vegetables“ in texture analysis)
Line 109: vegetables (????)...radish (????)
Lines 114-115: The sentence is not well worded.
Lines 115-119: These sentences are not well worded.
Line 133: replace „flavor“ with „volatile components“
Lines 136-137: „The alteration of texture, color,...“ is an unclear sentence; it should be reformulated.
Lines 149-150: „(including radish and green vegetables)“ – unclear part of sentence...
Line 154: „Determination of sensory evaluation“ should be replaced with „Sensory evaluation“ or „Determination of sensory properties“
Line 180: J/(mol k) should be replaced with J/(mol K)
Line 207: „Single-way“ should be replaced with „One-way“
Line 213: „flavors“ shoud be replaced with „tastes“
RESULTS AND DISCUSSION section
Lines 219-235 (and Table 3): Why the changes in texture at the storage temperature of 25 °C are not shown? vegetables (????)...radish (????)
Lines 236-254 (and Table 4): Why the color changes at the storage temperature of 25 °C are not shown? vegetables (????)...radish (????)
Lines 255-265 (and Figure 1): Why the sensory evaluation of samples storaged at 25 °C are not shown?
Line 261: Instead „49 and 156 days“, it should stand „126 and 49 days“. (See Figure 1)
Lines 273-275 and Lines 284-286: What is the cause of increasing total acids content during storage??
Line 281: „D-isoascorbate sodium“ should be replaced with „sodium D-isoascorbate“.
Line 287: Maybe it is better to replace the word „flavor“ with the word „volatiles“.
Lines 289-290: Reformulate the sentence „GC-MS mainly measures the change of smell...“ It is better to say: „GC-MS mainly measures the change of volatiles that affect smell...“
Line 291: correct „flacor“ for „flavor“, that is „aroma compounds“
Table 5: .....In the title of Table 5 – unclear meaning of „preliminary identified“ ?????? Also, Dimethyl disulfide, Dimethyl tetrasulfide and Di-2-propenyl trisulfide are not aldehydes..
Line 292: „3 acids“...In the Table 5 there are no acids (At the bottom of the Table 5 on page 22, sorbic acid, octanoic acid and decanoic acid are incorrectly classified as phenolic compounds).
Line 316: The mentioned Figure 4 does not exist in the attachment .
Lines 344-352: The results of the NIR analysis should be discussed in more detail.
ADDITIONAL COMMENTS
The objective of the paper titled „Shelf life prediction and critical value of quality index of Sichuan sauerkraut based on kinetic model and principal component analysis“ according to the authors, was to study effect of storage temperature on the quality of Sichuan sauerkraut.
In my opinion, the objectives of this work are interesting. Nevertheless the results of this research are very useful for Sichuan sauerkraut supply chain, but the paper in some aspects must be improved to be publishable. It should be explained in more detail (in the Introduction section) at which temperatures this product is usually stored, ie why such high storage temperatures (35 °C and 45 °C) were chosen in the research. Also, the results (section Results and discussion) describing changes in quality at a "normal storage temperature" of 25 °C are omitted. Especially because of the fact that in the sections Abstract and Material and Methods, it is stated that the changes will be monitored at 25 °C, as well. Unfortunately, the results describing the changes at this temperature (25 °C) are missing both in the text and in Figures 1-7 and Tables 3-6. Since we assume that the most common storage temperature of Sichuan sauerkraut is 25 °C, these results should be shown as standard, or the paper should show the results for only 35 and 45 °C. Also, English should be improved by a native speaker.
Author Response
RESPONSES TO Reviewer 1:
ABSTRACT section:
- Lines 21-23: “The alteration of texture, color,...” is an unclear sentence; it should be reformulated.
Response: Thank you very much for your suggestions. According to the suggestion, we have changed the sentence “The alteration of texture, color, total acid, microorganism, near infrared, volatile components, taste and sensory evaluation were investigated during the storage of Sichuan sauerkraut under 25, 35 and 45 ℃ respectively.” to “The texture, color, total acid, microbe, near infrared, flavor, taste and sensory evaluation of Sichuan sauerkraut during storage at 25℃, 35℃ and 45℃ were studied.”. Please see the revised lines 21-23.
- 2. Line 22: replace “flavor” with “volatile components”
Response: Thank you very much for your suggestions. we have replaced “flavor” with “volatile components”. Please see the revised line 22.
- 3. Lines 24-27: “Principal component analysis...” is also unclear sentence; it should be reformulated.
Response: Thank you very much for your suggestions. According to the suggestion, we have changed the sentence “Principal component analysis (PCA) and Fisher discriminant analysis (FDA) were used to process the e-tongue data, and the Sichuan sauerkraut with different storage time could be divided into three types: completely acceptable period, acceptable period and unacceptable period.” to “Principal component analysis (PCA) and Fisher discriminant analysis (FDA) were used to analyze the electronic tongue data. According to the above analysis, Sichuan sauerkraut with different storage time can be divided into three types: completely acceptable period, acceptable period and unacceptable period.”. Please see the revised lines 23-27.
- 4.Lines 35-37: “The establishment of the model...” is also unclear sentence; it should be reformulated.
Response: Thank you very much for your suggestions. According to the suggestion, we have changed the sentence “The establishment of the model can provide dealers and consumers with the shelf life and edible standards of Sichuan sauerkraut.” to “It is a better choice for dealers and consumers to judge the shelf life and edibility of food by shelf life model.”. Please see the revised lines 35-36.
INTRODUCTION section:
- 5. Lines 48-58: These sentences are not clear enough and need to be reformulated.
Response: Thank you very much for your suggestions. According to the suggestion, we have changed the sentence “In the production process of …” to “In the production process of Sichuan sauerkraut, heat sterilization technology was used to inactivate most of the microorganisms. The addition of potassium sorbate and sodium D-isoascorbate inhibited the growth of lactic acid bacteria and molds. Therefore, microorganisms can not be used as a standard to judge the shelf life of modified products. From the perspective of consumers, it is more reasonable to judge the shelf life of sauerkraut by volatile components index. In practical application, the shelf life model is of great significance for distributors and consumers to determine the shelf life and the best use period of food.”. Please see the revised lines 48-57.
- 6. Lines 59-63: These sentences are not clear enough and need to be reformulated.
Response: Thank you very much for your suggestions. According to the suggestion, we have changed the sentence “For food that needs to be stored…” to “For the food that needs to be stored, the dynamic model can reduce the time cost in the storage process and improve the accuracy of predicting the quality index and shelf life of the food. Most foods contain a lot of nutrients and are very sensitive to temperature, so they are often in a state of high entropy and low enthalpy during storage. The shelf-life model can represent the quality changes of food during storage.”. Please see the revised lines 71-76.
- 7. In general, the Introduction section should be revised in terms of language. Also, it is not clear why the authors have chosen these high storage temperatures. Are these storage temeperatures of Sichuan sauerkraut common in practice, or in any part of the supply chain? This should be described more precisely.
Response: Thank you very much for your feedback and comments on our manuscript.
According to the suggestion, We have revised the Introduction section. Storage at 35°C and 45°C is reasonable and shelf life experiments can be performed. The reason for using reasonable high temperature storage is that a more complete model can be established by accelerated experiments.In practice, Sichuan sauerkraut is stored at room temperature. After Sichuan sauerkraut is produced, in order to determine the shelf life of the product, it is usually determined by placing it at room temperature for a certain period of time. From a time cost perspective, this is a time-consuming practice. Therefore, it is necessary to establish a shelf life prediction model through accelerated experiments, which can reduce a lot of time costs. Please see the revised lines 53-81 and 87-91.
“accelerated experiments” references:
Jafari, S.M.; Ganje, M.; Dehnad, D.; Ghanbari, V.; Hajitabar, J. Arrhenius equation modeling for the shelf life prediction of tomato paste containing a natural preservative. J Sci Food Agric.
Zi-Chun, W.; Yu-Xi, Y.; Huan-Ping, A.; Hao, Y.; Di-Feng, R.; Jun, L. The shelf-life of chestnut rose beverage packaged in PEN/PET bottles under long term storage: a comparison to packaging in ordinary PET bottles. Food Chemistry 2022, 370.
Yan, S.; Ting, L.; Xiao-Yang, L.; Man-Tong, Z.; Fa-Wen, Y.; Kanyasiri, R.; Da-Yong, Z. Improving the oxidative stability and lengthening the shelf life of DHA algae oil with composite antioxidants. Food Chemistry 2020, 313, 126139-126139.
Jiang, Y.; Yang, X.; Jin, H.; Feng, X.; Tian, F.; Song, Y.; Ren, Y.; Man, C.; Zhang, W. Shelf-life prediction and chemical characteristics analysis of milk formula during storage. Lwt-Food Science and Technology 2021, 144.
MATERIALS AND METHODS section
- Line 97: Radish (??????). It should be better explained (what is the meaning of terms “radish” and “vegetables” in texture analysis). According to the title, the research is aimed at studying the change in the quality of sauerkraut, not radish or vegetables.
Response: Thank you very much for your suggestions. It was our oversight to not state the raw materials used in this product. The materials used in this Sichuan sauerkraut product are radish and green cabbage, so the objects measured in the texture analysis are these two materials. Please see the revised lines 99-100.
- Line 99: Vegetable (??????). It should be better explained (what is the meaning of terms „radish“ and „vegetables“ in texture analysis)
Response: Thank you very much for your suggestions. Since this question is consistent with the above answer, we do not repeat the explanation again. Please see the revised lines 99-100.
- Line 109: vegetables (????)...radish (????)
Response: Thank you very much for your suggestions. Since this question is consistent with the above answer, we do not repeat the explanation again. Please see the revised lines 99-100.
- 1 Lines 114-115: The sentence is not well worded.
Response: Thank you very much for your suggestions. According to the suggestion, we have changed the sentence “Approximately 25 g homogenous sample was mixed with 50 mL deionized water removed CO2 under boiling for 30 min.” to “Deionized water was heated to a boiling state for 15 minutes to remove CO2, and then cooled to 70-80°C. A 25g sample was crushed in a beater. The sample was then mixed with deionized water, placed in conical flask with condenser, and boiled for 30 min.”. Please see the revised lines 124-127.
- 1Lines 115-119: These sentences are not well worded.
Response: Thank you very much for your suggestions. According to the suggestion, we have changed the sentence “After cooling, the solution was…” to “The sample was taken out, and after cooling to room temperature, the filtrate was collected by filtration with fast filter paper. Take 25 mL of test solution and put it in a beaker; put it on a magnetic stirrer, immerse the electrode of the acidity meter, and start the stirrer; quickly titrate with 0.1mol/L NaOH solution to pH 8.20. When calculating the total acid content, use the conversion factor for lactic acid.”. Please see the revised lines 127-132.
- 1 Line 133: replace “flavor” with “volatile components”.
Response: Thank you very much for your suggestions. we have replaced “flavor” with “volatile components”. Please see the revised line 145.
- 1Lines 136-137: “The alteration of texture, color,...” is an unclear sentence; it should be reformulated.
Response: Thank you very much for your suggestions. According to the suggestion, we have changed the sentence “The sample cut and 3.0 g of chopped sample were placed in a 15 mL vial.” to “A 3.0 g sample was placed in a 15 mL gassed bottle.”. Please see the revised lines 148-149.
- 1Lines 149-150: “(including radish and green vegetables)” – unclear part of sentence...
Response: Thank you very much for your suggestions. According to the suggestion, we have changed the sentence “Twenty grams of Sichuan sauerkraut sample…” to “20 g of radish and green vegetables samples were taken for near-infrared detection.”. Please see the revised lines 161-163.
- 1Line 154: “Determination of sensory evaluation” should be replaced with “Sensory evaluation” or “Determination of sensory properties”.
Response: Thank you very much for your suggestions. we have replaced “Determination of sensory evaluation” with “Sensory evaluation”. Please see the revised line 166.
- 1Line 180: J/(mol k) should be replaced with J/(mol K)
Response: Thank you very much for your suggestions. we have replaced “J/(mol k)” with “J/(mol K)”. Please see the revised line 192.
- 1Line 207: “Single-way” should be replaced with “One-way”.
Response: Thank you very much for your suggestions. we have replaced “Single-way” with “One-way”. Please see the revised line 219.
- 1Line 213: “flavors” shoud be replaced with “tastes”.
Response: Thank you very much for your suggestions. we have replaced “flavors” with “tastes”. Please see the revised line 225.
RESULTS AND DISCUSSION section
- Lines 219-235 (and Table 3): Why the changes in texture at the storage temperature of 25 °C are not shown? vegetables (????)...radish (????)
Response: Thank you very much for your suggestions. We have added information about the texture change of Sichuan sauerkraut at 25°C. The analysis of the change in texture at 25°C is as follows: At 25 ℃, the hardness and chewiness of radish ranged from 763.28 to 182.95 gf, 93.04 to 25.52 gf, respectively. The hardness and chewiness of green vegetables ranged from 277.87 to 76.22 gf and 66.15 to 27.12 gf, respectively. Please see the revised lines 241-244 and Table 3.
- 2 Lines 236-254 (and Table 4): Why the color changes at the storage temperature of 25 °C are not shown? vegetables (????)...radish (????)
Response: Thank you very much for your suggestions. We have added information about the color change of Sichuan sauerkraut at 25°C. The analysis of the change in color at 25°C is as follows: The Sichuan sauerkraut stored at 25 ℃, L* value of radish in Sichuan sauerkraut varied from 58.57 to 44.79, a* value varied from -3.06 to 2.89, b* value varied from 18.84 to 14.47, ΔE value varied from 18.77 to 15.97; Vegetables showed L* values was from 49.22 to 44.98, a* value was from -4.98 to -1.39, b* value was from 21.68 to 15.22, ΔE value from 4.86 to 8.75. Please see the revised lines 265-270 and Table 4(C).
- 2 Lines 255-265 (and Figure 1): Why the sensory evaluation of samples storaged at 25 °C are not shown?
Response: Thank you very much for your suggestions. We have added information about the sensory evaluation of Sichuan sauerkraut at 25°C. The analysis of the sensory evaluation at 25°C is as follows: Sichuan sauerkraut is unacceptable at 25 ℃ when the storage time reached 336 days. Please see the revised lines 281-282 and Table Fig. 1(c).
- Line 261: Instead “49 and 156 days”, it should stand “126 and 49 days”. (See Figure 1)
Response: Thank you very much for your suggestions. we have replaced “49 and 156 days” with “126 and 49 days”. Please see the revised line 281-282.
- 2 Lines 273-275 and Lines 284-286: What is the cause of increasing total acids content during storage??
Response: Thank you very much for your suggestions. After our analysis, we found that although the number of microorganisms in Sichuan sauerkraut is very less, it has a certain impact on the quality of sauerkraut. This is due to the anaerobic environment lactic acid bacteria and some other microorganisms will consume sugars and increase the total acid content. With the increase of total acid content and high salt environment, the growth of microorganisms will be inhibited. Therefore, the microorganisms showed a trend of first rising and then falling during the storage process. Please see the revised lines 295-297 and 306-311.
“the cause of increasing total acids content during storage” references:
Xiong, T.; Li, J.B.; Liang, F.; Wang, Y.P.; Guan, Q.Q.; Xie, M.Y. Effects of salt concentration on Chinese sauerkraut fermentation. Lwt-Food Science and Technology 2016, 69, 169-174.
Du, R.; Song, G.; Zhao, D.; Sun, J.; Ping, W.; Ge, J. Lactobacillus casei starter culture improves vitamin content, increases acidity and decreases nitrite concentration during sauerkraut fermentation. International Journal of Food Science and Technology 2018, 53.
- 2Line 281: “D-isoascorbate sodium” should be replaced with “sodium D-isoascorbate”.
Response: Thank you very much for your suggestions. we have replaced “D-isoascorbate sodium” with “sodium D-isoascorbate”. Please see the revised line 303.
- 2Line 287: Maybe it is better to replace the word “flavor” with the word “volatiles”.
Response: Thank you very much for your suggestions. we have replaced “flavor” with “volatiles”. Please see the revised line 312.
- 2Lines 289-290: Reformulate the sentence “GC-MS mainly measures the change of smell...” It is better to say: “GC-MS mainly measures the change of volatiles that affect smell...”.
Response: Thank you very much for your suggestions. According to the suggestion, we have changed the sentence “GC-MS mainly measures the change of smell...” to “GC-MS mainly measures the change of volatiles that affect smell...”. Please see the revised lines 314-315.
- 2Line 291: correct “flacor” for “flavor”, that is “aroma compounds”.
Response: Thank you very much for your suggestions. we have replaced “flacor” with “flavor”. Please see the revised line 316.
- 2 Table 5: .....In the title of Table 5 – unclear meaning of “preliminary identified” ?????? Also, Dimethyl disulfide, Dimethyl tetrasulfide and Di-2-propenyl trisulfide are not aldehydes.
Response: Thank you very much for your feedback and comments on our manuscript. According to the suggestion, we have changed the title “The volatile components of Sichuan sauerkraut during storage (45 °C, 35 ℃) were preliminarily identified” to “Changes of volatile components in Sichuan sauerkraut during storage (45℃, 35℃)”. And we classified Dimethyl disulfide, Dimethyl tetrasulfide and Di-2-propenyl trisulfide into thioether compounds. Please see the revised line 317-320 and Table 5.
- Line 292: „3 acids“...In the Table 5 there are no acids (At the bottom of the Table 5 on page 22, sorbic acid, octanoic acid and decanoic acid are incorrectly classified as phenolic compounds).
Response: Thank you very much for your feedback and comments on our manuscript. According to the suggestion, We classified sorbic acid, octanoic acid and decanoic acid into acid compounds. Please see the revised Table 5.
- Line 316: The mentioned Figure 4 does not exist in the attachment .
Response: Thank you very much for your suggestions. We did not clearly indicate the exact location of Figure 4, which has been changed first. we have replaced “Fig. 4” with “Fig. 4(a) and Fig 4(b)”. Please see the revised line 350.
- 3 Lines 344-352: The results of the NIR analysis should be discussed in more detail.
Response: Thank you very much for your suggestions. Since NIR data are used to establish BP-ANN model, there are few analyses on this part. Here's what was added according to the suggestion: With the increase of storage time, the intensity of the three spectral absorption peaks increased. This is due to the breakdown of the sauerkraut cell wall resulting in water loss. Please see the revised lines 386-388.
ADDITIONAL COMMENTS
- 3 In my opinion, the objectives of this work are interesting. Nevertheless the results of this research are very useful for Sichuan sauerkraut supply chain, but the paper in some aspects must be improved to be publishable. It should be explained in more detail (in the Introduction section) at which temperatures this product is usually stored, ie why such high storage temperatures (35 °C and 45 °C) were chosen in the research. Also, the results (section Results and discussion) describing changes in quality at a "normal storage temperature" of 25 °C are omitted. Especially because of the fact that in the sections Abstract and Material and Methods, it is stated that the changes will be monitored at 25 °C, as well. Unfortunately, the results describing the changes at this temperature (25 °C) are missing both in the text and in Figures 1-7 and Tables 3-6. Since we assume that the most common storage temperature of Sichuan sauerkraut is 25 °C, these results should be shown as standard, or the paper should show the results for only 35 and 45 °C. Also, English should be improved by a native speaker.
Response: Thank you very much for your comments on our manuscript. Your comments have been of great help to us. We have revised the paper accordingly according to your comments. You mentioned in the previous data about texture, color and sensory evaluation at 25℃, we have made corresponding supplements. This makes the paper more convincing. For other indicators, we do not repeat them. This is because it is already fully covered in the modeling section. Thank you again for your help with this paper.
Reviewer 2 Report
The manuscript "Shelf life prediction and critical value of quality index of Sichuan sauerkraut based on kinetic model and principal component analysis" is well written and contains information relevant to the field.
The main objective of the article is to evaluate the shelf life of Sichuan sauerkraut, which undergoes a series of changes during its storage. To achieve this objective, the authors carried out the storage at three temperatures (25, 35 and 45 °C) and analyzed parameters such as texture, color, total acidity, microorganisms, near infrared and sensory evaluation.
I believe the article is interesting, as there are few manuscripts in the literature on storing Sichuan sauerkraut at different temperatures, in addition to the authors using refined techniques such as E-tongue, in addition to the experimental design being analyzed by PCA , in addition to mathematical models.
Compared with other published material, in addition to the techniques used, the analysis and way of presenting the results has a great differential.
And, the manuscript is well written and presents relevant results, in addition to evaluating the product for 154 days. And the order of presentation makes the manuscript easier to read and understand.
Overall, the authors obtained the main answer of the work, however the conclusion is very extensive, with a lot of information already presented previously, I believe that the authors could reduce to present only what is necessary, according to the main objective.
Here are small considerations for improvement:
- Display more information about the "Determination of taste" analysis
- The conclusion must be summarized
Author Response
RESPONSES TO Reviewer 2:
Comments and Suggestions:
- Display more information about the "Determination of taste" analysis.
Response: Thank you very much for your comments on our manuscript. According to the suggestions, we have supplemented the analysis of "determination of taste", as follows: It can be seen from the figure that the astringency, aftertaste-A, richness and saltiness of Sichuan sauerkraut did not change significantly during storage. This shows that these flavors are not the main reason for sauerkraut. The changes in sourness and umami were more obvious, which was consistent with the results of sensory evaluation. The sourness and umami of Sichuan sauerkraut were 14.79 to 17.18, 2.03 to 1.31 respectively when stored at 45℃; at 35℃, the sourness and umami were respectively 14.79 to 18.23, 2.03 to 1.40; at 25℃, the sourness and umami were respectively 14.79 to 18.33, 2.03 to 1.43. Please see the revised lines 331-340 and Fig. S1.
- The conclusion must be summarized.
Response: Thank you very much for your comments on our manuscript. According to the suggestions, we supplemented the conclusions as follows: It is feasible to store Sichuan sauerkraut by setting the temperature at 25°C, 35°C and 45°C through accelerated experiments. All kinds of data measured by this scheme can be well applied to Arrhenius and BP-ANN models. The reliability of the above model is proved by verifying the prediction accuracy of the model. Such models are of great significance in the production, transportation and sales of Sichuan sauerkraut. Thank you again for your help with our paper. Please see the revised lines 488-493.
Round 2
Reviewer 1 Report
The authors successfully responded to the reviewer's remarks.
Author Response
ABSTRACT section:
- Lines 21-23: “The alteration of texture, color,...” is an unclear sentence; it should be reformulated.
Response: Thank you very much for your suggestions. According to the suggestion, we have changed the sentence “The alteration of texture, color, total acid, microorganism, near infrared, volatile components, taste and sensory evaluation were investigated during the storage of Sichuan sauerkraut under 25, 35 and 45 ℃ respectively.” to “The texture, color, total acid, microbe, near infrared, flavor, taste and sensory evaluation of Sichuan sauerkraut during storage at 25℃, 35℃ and 45℃ were studied.”. Please see the revised lines 21-23.
- Line 22: replace “flavor” with “volatile components”
Response: Thank you very much for your suggestions. we have replaced “flavor” with “volatile components”. Please see the revised line 22.
- Lines 24-27: “Principal component analysis...” is also unclear sentence; it should be reformulated.
Response: Thank you very much for your suggestions. According to the suggestion, we have changed the sentence “Principal component analysis (PCA) and Fisher discriminant analysis (FDA) were used to process the e-tongue data, and the Sichuan sauerkraut with different storage time could be divided into three types: completely acceptable period, acceptable period and unacceptable period.” to “Principal component analysis (PCA) and Fisher discriminant analysis (FDA) were used to analyze the electronic tongue data. According to the above analysis, Sichuan sauerkraut with different storage time can be divided into three types: completely acceptable period, acceptable period and unacceptable period.”. Please see the revised lines 23-27.
- Lines 35-37: “The establishment of the model...” is also unclear sentence; it should be reformulated.
Response: Thank you very much for your suggestions. According to the suggestion, we have changed the sentence “The establishment of the model can provide dealers and consumers with the shelf life and edible standards of Sichuan sauerkraut.” to “It is a better choice for dealers and consumers to judge the shelf life and edibility of food by shelf life model.”. Please see the revised lines 35-36.
INTRODUCTION section:
- Lines 48-58: These sentences are not clear enough and need to be reformulated.
Response: Thank you very much for your suggestions. According to the suggestion, we have changed the sentence “In the production process of …” to “In the production process of Sichuan sauerkraut, heat sterilization technology was used to inactivate most of the microorganisms. The addition of potassium sorbate and sodium D-isoascorbate inhibited the growth of lactic acid bacteria and molds. Therefore, microorganisms can not be used as a standard to judge the shelf life of modified products. From the perspective of consumers, it is more reasonable to judge the shelf life of sauerkraut by volatile components index. In practical application, the shelf life model is of great significance for distributors and consumers to determine the shelf life and the best use period of food.”. Please see the revised lines 48-57.
- Lines 59-63: These sentences are not clear enough and need to be reformulated.
Response: Thank you very much for your suggestions. According to the suggestion, we have changed the sentence “For food that needs to be stored…” to “For the food that needs to be stored, the dynamic model can reduce the time cost in the storage process and improve the accuracy of predicting the quality index and shelf life of the food. Most foods contain a lot of nutrients and are very sensitive to temperature, so they are often in a state of high entropy and low enthalpy during storage. The shelf-life model can represent the quality changes of food during storage.”. Please see the revised lines 71-76.
- In general, the Introduction section should be revised in terms of language. Also, it is not clear why the authors have chosen these high storage temperatures. Are these storage temeperatures of Sichuan sauerkraut common in practice, or in any part of the supply chain? This should be described more precisely.
Response: Thank you very much for your feedback and comments on our manuscript.
According to the suggestion, We have revised the Introduction section. Storage at 35°C and 45°C is reasonable and shelf life experiments can be performed. The reason for using reasonable high temperature storage is that a more complete model can be established by accelerated experiments.In practice, Sichuan sauerkraut is stored at room temperature. After Sichuan sauerkraut is produced, in order to determine the shelf life of the product, it is usually determined by placing it at room temperature for a certain period of time. From a time cost perspective, this is a time-consuming practice. Therefore, it is necessary to establish a shelf life prediction model through accelerated experiments, which can reduce a lot of time costs. Please see the revised lines 53-81 and 87-91.
“accelerated experiments” references:
Jafari, S.M.; Ganje, M.; Dehnad, D.; Ghanbari, V.; Hajitabar, J. Arrhenius equation modeling for the shelf life prediction of tomato paste containing a natural preservative. J Sci Food Agric.
Zi-Chun, W.; Yu-Xi, Y.; Huan-Ping, A.; Hao, Y.; Di-Feng, R.; Jun, L. The shelf-life of chestnut rose beverage packaged in PEN/PET bottles under long term storage: a comparison to packaging in ordinary PET bottles. Food Chemistry 2022, 370.
Yan, S.; Ting, L.; Xiao-Yang, L.; Man-Tong, Z.; Fa-Wen, Y.; Kanyasiri, R.; Da-Yong, Z. Improving the oxidative stability and lengthening the shelf life of DHA algae oil with composite antioxidants. Food Chemistry 2020, 313, 126139-126139.
Jiang, Y.; Yang, X.; Jin, H.; Feng, X.; Tian, F.; Song, Y.; Ren, Y.; Man, C.; Zhang, W. Shelf-life prediction and chemical characteristics analysis of milk formula during storage. Lwt-Food Science and Technology 2021, 144.
MATERIALS AND METHODS section
- Line 97: Radish (??????). It should be better explained (what is the meaning of terms “radish” and “vegetables” in texture analysis). According to the title, the research is aimed at studying the change in the quality of sauerkraut, not radish or vegetables.
Response: Thank you very much for your suggestions. It was our oversight to not state the raw materials used in this product. The materials used in this Sichuan sauerkraut product are radish and green cabbage, so the objects measured in the texture analysis are these two materials. Please see the revised lines 99-100.
- Line 99: Vegetable (??????). It should be better explained (what is the meaning of terms „radish“ and „vegetables“ in texture analysis)
Response: Thank you very much for your suggestions. Since this question is consistent with the above answer, we do not repeat the explanation again. Please see the revised lines 99-100.
- Line 109: vegetables (????)...radish (????)
Response: Thank you very much for your suggestions. Since this question is consistent with the above answer, we do not repeat the explanation again. Please see the revised lines 99-100.
- 1 Lines 114-115: The sentence is not well worded.
Response: Thank you very much for your suggestions. According to the suggestion, we have changed the sentence “Approximately 25 g homogenous sample was mixed with 50 mL deionized water removed CO2 under boiling for 30 min.” to “Deionized water was heated to a boiling state for 15 minutes to remove CO2, and then cooled to 70-80°C. A 25g sample was crushed in a beater. The sample was then mixed with deionized water, placed in conical flask with condenser, and boiled for 30 min.”. Please see the revised lines 124-127.
- 1Lines 115-119: These sentences are not well worded.
Response: Thank you very much for your suggestions. According to the suggestion, we have changed the sentence “After cooling, the solution was…” to “The sample was taken out, and after cooling to room temperature, the filtrate was collected by filtration with fast filter paper. Take 25 mL of test solution and put it in a beaker; put it on a magnetic stirrer, immerse the electrode of the acidity meter, and start the stirrer; quickly titrate with 0.1mol/L NaOH solution to pH 8.20. When calculating the total acid content, use the conversion factor for lactic acid.”. Please see the revised lines 127-132.
- 1 Line 133: replace “flavor” with “volatile components”.
Response: Thank you very much for your suggestions. we have replaced “flavor” with “volatile components”. Please see the revised line 145.
- 1Lines 136-137: “The alteration of texture, color,...” is an unclear sentence; it should be reformulated.
Response: Thank you very much for your suggestions. According to the suggestion, we have changed the sentence “The sample cut and 3.0 g of chopped sample were placed in a 15 mL vial.” to “A 3.0 g sample was placed in a 15 mL gassed bottle.”. Please see the revised lines 148-149.
- 1Lines 149-150: “(including radish and green vegetables)” – unclear part of sentence...
Response: Thank you very much for your suggestions. According to the suggestion, we have changed the sentence “Twenty grams of Sichuan sauerkraut sample…” to “20 g of radish and green vegetables samples were taken for near-infrared detection.”. Please see the revised lines 161-163.
- 1Line 154: “Determination of sensory evaluation” should be replaced with “Sensory evaluation” or “Determination of sensory properties”.
Response: Thank you very much for your suggestions. we have replaced “Determination of sensory evaluation” with “Sensory evaluation”. Please see the revised line 166.
- 1Line 180: J/(mol k) should be replaced with J/(mol K)
Response: Thank you very much for your suggestions. we have replaced “J/(mol k)” with “J/(mol K)”. Please see the revised line 192.
- 1Line 207: “Single-way” should be replaced with “One-way”.
Response: Thank you very much for your suggestions. we have replaced “Single-way” with “One-way”. Please see the revised line 219.
- 1Line 213: “flavors” shoud be replaced with “tastes”.
Response: Thank you very much for your suggestions. we have replaced “flavors” with “tastes”. Please see the revised line 225.
RESULTS AND DISCUSSION section
- Lines 219-235 (and Table 3): Why the changes in texture at the storage temperature of 25 °C are not shown? vegetables (????)...radish (????)
Response: Thank you very much for your suggestions. We have added information about the texture change of Sichuan sauerkraut at 25°C. The analysis of the change in texture at 25°C is as follows: At 25 ℃, the hardness and chewiness of radish ranged from 763.28 to 182.95 gf, 93.04 to 25.52 gf, respectively. The hardness and chewiness of green vegetables ranged from 277.87 to 76.22 gf and 66.15 to 27.12 gf, respectively. Please see the revised lines 241-244 and Table 3.
- 2 Lines 236-254 (and Table 4): Why the color changes at the storage temperature of 25 °C are not shown? vegetables (????)...radish (????)
Response: Thank you very much for your suggestions. We have added information about the color change of Sichuan sauerkraut at 25°C. The analysis of the change in color at 25°C is as follows: The Sichuan sauerkraut stored at 25 ℃, L* value of radish in Sichuan sauerkraut varied from 58.57 to 44.79, a* value varied from -3.06 to 2.89, b* value varied from 18.84 to 14.47, ΔE value varied from 18.77 to 15.97; Vegetables showed L* values was from 49.22 to 44.98, a* value was from -4.98 to -1.39, b* value was from 21.68 to 15.22, ΔE value from 4.86 to 8.75. Please see the revised lines 265-270 and Table 4(C).
- 2 Lines 255-265 (and Figure 1): Why the sensory evaluation of samples storaged at 25 °C are not shown?
Response: Thank you very much for your suggestions. We have added information about the sensory evaluation of Sichuan sauerkraut at 25°C. The analysis of the sensory evaluation at 25°C is as follows: Sichuan sauerkraut is unacceptable at 25 ℃ when the storage time reached 336 days. Please see the revised lines 281-282 and Table Fig. 1(c).
- Line 261: Instead “49 and 156 days”, it should stand “126 and 49 days”. (See Figure 1)
Response: Thank you very much for your suggestions. we have replaced “49 and 156 days” with “126 and 49 days”. Please see the revised line 281-282.
- 2 Lines 273-275 and Lines 284-286: What is the cause of increasing total acids content during storage??
Response: Thank you very much for your suggestions. After our analysis, we found that although the number of microorganisms in Sichuan sauerkraut is very less, it has a certain impact on the quality of sauerkraut. This is due to the anaerobic environment lactic acid bacteria and some other microorganisms will consume sugars and increase the total acid content. With the increase of total acid content and high salt environment, the growth of microorganisms will be inhibited. Therefore, the microorganisms showed a trend of first rising and then falling during the storage process. Please see the revised lines 295-297 and 306-311.
“the cause of increasing total acids content during storage” references:
Xiong, T.; Li, J.B.; Liang, F.; Wang, Y.P.; Guan, Q.Q.; Xie, M.Y. Effects of salt concentration on Chinese sauerkraut fermentation. Lwt-Food Science and Technology 2016, 69, 169-174.
Du, R.; Song, G.; Zhao, D.; Sun, J.; Ping, W.; Ge, J. Lactobacillus casei starter culture improves vitamin content, increases acidity and decreases nitrite concentration during sauerkraut fermentation. International Journal of Food Science and Technology 2018, 53.
- 2Line 281: “D-isoascorbate sodium” should be replaced with “sodium D-isoascorbate”.
Response: Thank you very much for your suggestions. we have replaced “D-isoascorbate sodium” with “sodium D-isoascorbate”. Please see the revised line 303.
- 2Line 287: Maybe it is better to replace the word “flavor” with the word “volatiles”.
Response: Thank you very much for your suggestions. we have replaced “flavor” with “volatiles”. Please see the revised line 312.
- 2Lines 289-290: Reformulate the sentence “GC-MS mainly measures the change of smell...” It is better to say: “GC-MS mainly measures the change of volatiles that affect smell...”.
Response: Thank you very much for your suggestions. According to the suggestion, we have changed the sentence “GC-MS mainly measures the change of smell...” to “GC-MS mainly measures the change of volatiles that affect smell...”. Please see the revised lines 314-315.
- 2Line 291: correct “flacor” for “flavor”, that is “aroma compounds”.
Response: Thank you very much for your suggestions. we have replaced “flacor” with “flavor”. Please see the revised line 316.
- 2 Table 5: .....In the title of Table 5 – unclear meaning of “preliminary identified” ?????? Also, Dimethyl disulfide, Dimethyl tetrasulfide and Di-2-propenyl trisulfide are not aldehydes.
Response: Thank you very much for your feedback and comments on our manuscript. According to the suggestion, we have changed the title “The volatile components of Sichuan sauerkraut during storage (45 °C, 35 ℃) were preliminarily identified” to “Changes of volatile components in Sichuan sauerkraut during storage (45℃, 35℃)”. And we classified Dimethyl disulfide, Dimethyl tetrasulfide and Di-2-propenyl trisulfide into thioether compounds. Please see the revised line 317-320 and Table 5.
- Line 292: „3 acids“...In the Table 5 there are no acids (At the bottom of the Table 5 on page 22, sorbic acid, octanoic acid and decanoic acid are incorrectly classified as phenolic compounds).
Response: Thank you very much for your feedback and comments on our manuscript. According to the suggestion, We classified sorbic acid, octanoic acid and decanoic acid into acid compounds. Please see the revised Table 5.
- Line 316: The mentioned Figure 4 does not exist in the attachment .
Response: Thank you very much for your suggestions. We did not clearly indicate the exact location of Figure 4, which has been changed first. we have replaced “Fig. 4” with “Fig. 4(a) and Fig 4(b)”. Please see the revised line 350.
- 3 Lines 344-352: The results of the NIR analysis should be discussed in more detail.
Response: Thank you very much for your suggestions. Since NIR data are used to establish BP-ANN model, there are few analyses on this part. Here's what was added according to the suggestion: With the increase of storage time, the intensity of the three spectral absorption peaks increased. This is due to the breakdown of the sauerkraut cell wall resulting in water loss. Please see the revised lines 386-388.
ADDITIONAL COMMENTS
- 3 In my opinion, the objectives of this work are interesting. Nevertheless the results of this research are very useful for Sichuan sauerkraut supply chain, but the paper in some aspects must be improved to be publishable. It should be explained in more detail (in the Introduction section) at which temperatures this product is usually stored, ie why such high storage temperatures (35 °C and 45 °C) were chosen in the research. Also, the results (section Results and discussion) describing changes in quality at a "normal storage temperature" of 25 °C are omitted. Especially because of the fact that in the sections Abstract and Material and Methods, it is stated that the changes will be monitored at 25 °C, as well. Unfortunately, the results describing the changes at this temperature (25 °C) are missing both in the text and in Figures 1-7 and Tables 3-6. Since we assume that the most common storage temperature of Sichuan sauerkraut is 25 °C, these results should be shown as standard, or the paper should show the results for only 35 and 45 °C. Also, English should be improved by a native speaker.
Response: Thank you very much for your comments on our manuscript. Your comments have been of great help to us. We have revised the paper accordingly according to your comments. You mentioned in the previous data about texture, color and sensory evaluation at 25℃, we have made corresponding supplements. This makes the paper more convincing. For other indicators, we do not repeat them. This is because it is already fully covered in the modeling section. Thank you again for your help with this paper.